# PDE-Refiner: Achieving Accurate Long Rollouts with Neural PDE Solvers

**Phillip Lippe**
Microsoft Research AI4Science*
phillip.lippe@googlemail.com

**Bastiaan S. Veeling**
Microsoft Research AI4Science

**Paris Perdikaris**
Microsoft Research AI4Science

**Richard E. Turner**
Microsoft Research AI4Science

**Johannes Brandstetter**
Microsoft Research AI4Science
brandstetter@ml.jku.at

## Abstract

Time-dependent partial differential equations (PDEs) are ubiquitous in science and engineering. Recently, mostly due to the high computational cost of traditional solution techniques, deep neural network based surrogates have gained increased interest. The practical utility of such neural PDE solvers relies on their ability to provide accurate, stable predictions over long time horizons, which is a notoriously hard problem. In this work, we present a large-scale analysis of common temporal rollout strategies, identifying the neglect of non-dominant spatial frequency information, often associated with high frequencies in PDE solutions, as the primary pitfall limiting stable, accurate rollout performance. Based on these insights, we draw inspiration from recent advances in diffusion models to introduce PDE-Refiner; a novel model class that enables more accurate modeling of all frequency components via a multistep refinement process. We validate PDE-Refiner on challenging benchmarks of complex fluid dynamics, demonstrating stable and accurate rollouts that consistently outperform state-of-the-art models, including neural, numerical, and hybrid neural-numerical architectures. We further demonstrate that PDE-Refiner greatly enhances data efficiency, since the denoising objective implicitly induces a novel form of spectral data augmentation. Finally, PDE-Refiner's connection to diffusion models enables an accurate and efficient assessment of the model's predictive uncertainty, allowing us to estimate when the surrogate becomes inaccurate.

## 1 Introduction

In recent years, mostly due to a rapidly growing interest in modeling partial differential equations (PDEs), deep neural network based PDE surrogates have gained significant momentum as a more computationally efficient solution methodology [10, 81]. Recent approaches can be broadly classified into three categories: (i) neural approaches that approximate the solution function of the underlying PDE [24, 66]; (ii) hybrid approaches, where neural networks either augment numerical solvers or replace parts of them [1, 2, 18, 33, 43, 79]; (iii) neural approaches in which the learned evolution operator maps the current state to a future state of the system [4, 7, 9, 21, 49, 54, 68, 87].

---

*Work done during internship at Microsoft Research; on leave from University of Amsterdam.

37th Conference on Neural Information Processing Systems (NeurIPS 2023).

Approaches (i) have had great success in modeling inverse and high-dimensional problems [37], whereas approaches (ii) and (iii) have started to advance fluid and weather modeling in two and three dimensions [21, 39, 43, 46, 51, 60, 63, 67, 76, 89]. These problems are usually described by complex time-dependent PDEs. Solving this class of PDEs over long time horizons presents fundamental challenges. Conventional numerical methods suffer accumulating approximation effects, which in the temporal solution step can be counteracted by implicit methods [23, 36]. Neural PDE solvers similarly struggle with the effects of accumulating noise, an inevitable consequence of autoregressively propagating the solutions of the underlying PDEs over time [9, 43, 59, 79]. Another critique of neural PDE solvers is that – besides very few exceptions, e.g., [33] – they lack convergence guarantees and predictive uncertainty modeling, i.e., estimates of how much to trust the predictions. Whereas the former is in general notoriously difficult to establish in the context of deep learning, the latter links to recent advances in probabilistic neural modeling [17, 28, 30, 42, 72, 75, 82], and, thus, opens the door for new families of uncertainty-aware neural PDE solvers. In summary, to the best of our understanding, the most important desiderata for current time-dependent neural PDE solvers comprise long-term accuracy, long-term stability, and the ability to quantify predictive uncertainty.

In this work, we analyze common temporal rollout strategies, including simple autoregressive unrolling with varying history input, the pushforward trick [9], invariance preservation [58], and the Markov Neural Operator [50]. We test temporal modeling by state-of-the-art neural operators such as modern U-Nets [22] and Fourier Neural Operators (FNOs) [49], and identify a shared pitfall in all these unrolling schemes: neural solvers consistently neglect components of the spatial frequency spectrum that have low amplitude. Although these frequencies have minimal immediate impact, they still impact long-term dynamics, ultimately resulting in a noticeable decline in rollout performance.

Based on these insights, we draw inspiration from recent advances in diffusion models [30, 61, 70] to introduce PDE-Refiner. PDE-Refiner is a novel model class that uses an iterative refinement process to obtain accurate predictions over the whole frequency spectrum. This is achieved by an adapted Gaussian denoising step that forces the network to focus on information from all frequency components equally at different amplitude levels. We demonstrate the effectiveness of PDE-Refiner on solving the 1D Kuramoto-Sivashinsky equation and the 2D Kolmogorov flow, a variant of the incompressible Navier-Stokes flow. On both PDEs, PDE-Refiner models the frequency spectrum much more accurately than the baselines, leading to a significant gain in accurate rollout time.

## 2 Challenges of Accurate Long Rollouts

**Partial Differential Equations**. In this work, we focus on time-dependent PDEs in one temporal dimension, i.e., $t \in [0, T]$, and possibly multiple spatial dimensions, i.e., $\boldsymbol{x} = [x_1, x_2, \ldots, x_m] \in \mathcal{X}$. Time-dependent PDEs relate solutions $\boldsymbol{u}(t, \boldsymbol{x}) : [0, T] \times \mathcal{X} \to \mathbb{R}^n$ and respective derivatives for all points in the domain, where $\boldsymbol{u}^0(\boldsymbol{x})$ are *initial conditions* at time $t = 0$ and $B[\boldsymbol{u}](t, \boldsymbol{x}) = 0$ are *boundary conditions* with boundary operator $B$ when $\boldsymbol{x}$ lies on the boundary $\partial \mathcal{X}$ of the domain. Such PDEs can be written in the form [9]:

$$\boldsymbol{u}_t = F(t, \boldsymbol{x}, \boldsymbol{u}, \boldsymbol{u}_{\boldsymbol{x}}, \boldsymbol{u}_{\boldsymbol{x}\boldsymbol{x}}, \ldots),  \tag{1}$$

where the notation $\boldsymbol{u}_t$ is shorthand for the partial derivative $\partial \boldsymbol{u}/\partial t$, while $\boldsymbol{u}_{\boldsymbol{x}}, \boldsymbol{u}_{\boldsymbol{x}\boldsymbol{x}}, \ldots$ denote the partial derivatives $\partial \boldsymbol{u}/\partial \boldsymbol{x}, \partial^2 \boldsymbol{u}/\partial \boldsymbol{x}^2$ and so on[2]. Operator learning [48, 49, 53–55] relates solutions $\boldsymbol{u} : \mathcal{X} \to \mathbb{R}^n, \boldsymbol{u}' : \mathcal{X}' \to \mathbb{R}^{n'}$ defined on different domains $\mathcal{X} \in \mathbb{R}^m, \mathcal{X}' \in \mathbb{R}^{m'}$ via operators $\mathcal{G}: \mathcal{G} : (\boldsymbol{u} \in \mathcal{U}) \to (\boldsymbol{u}' \in \mathcal{U}')$, where $\mathcal{U}$ and $\mathcal{U}'$ are the spaces of $\boldsymbol{u}$ and $\boldsymbol{u}'$, respectively. For time-dependent PDEs, an evolution operator can be used to compute the solution at time $t + \Delta t$ from time $t$ as

$$\boldsymbol{u}(t + \Delta t) = \mathcal{G}_t(\Delta t, \boldsymbol{u}(t)),  \tag{2}$$

where $\mathcal{G}_t : \mathbb{R}_{>0} \times \mathbb{R}^n \to \mathbb{R}^n$ is the temporal update. To obtain predictions over long time horizons, a temporal operator could either be directly trained for large $\Delta t$ or recursively applied with smaller time intervals. In practice, the predictions of learned operators deteriorate for large $\Delta t$, while autoregressive approaches are found to perform substantially better [22, 50, 86].

**Long Rollouts for Time-Dependent PDEs**. We start with showcasing the challenges of obtaining long, accurate rollouts for autoregressive neural PDE solvers on the working example of the 1D

---

[2]$\partial \boldsymbol{u}/\partial \boldsymbol{x}$ represents a $m \times n$ dimensional Jacobian matrix $\boldsymbol{J}$ with entries $\boldsymbol{J}_{ij} = \partial u_i/\partial x_j$.

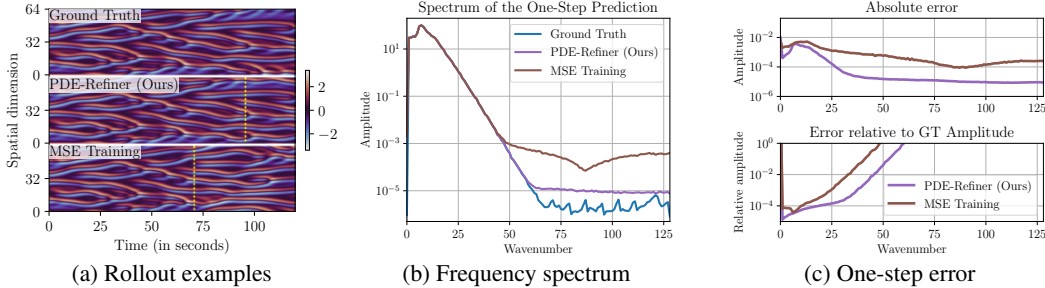

| (a) Rollout examples | (b) Frequency spectrum | (c) One-step error |

Figure 1: Challenges in achieving accurate long rollouts on the KS equation, comparing PDE-Refiner and an MSE-trained model. (a) Example trajectory with predicted rollouts. The yellow line indicates the time when the Pearson correlation between ground truth and prediction drops below 0.9. PDE-Refiner maintains an accurate rollout for longer than the MSE model. (b) Frequency spectrum over the spatial dimension of the ground truth data and one-step predictions. For PDE-Refiner, we show the average spectrum across 16 samples. (c) The spectra of the corresponding errors. The MSE model is only accurate for a small, high-amplitude frequency band, while PDE-Refiner supports a much larger frequency band, leading to longer accurate rollouts as in (a).

Kuramoto-Sivashinsky (KS) equation [44, 73]. The KS equation is a fourth-order nonlinear PDE, known for its rich dynamical characteristics and chaotic behavior [35, 40, 74]. It is defined as:

$$u_t + uu_x + u_{xx} + \nu u_{xxxx} = 0, \text{ }^3 \tag{3}$$

where $\nu$ is a viscosity parameter which we commonly set to $\nu = 1$. The nonlinear term $uu_x$ and the fourth-order derivative $u_{xxxx}$ make the PDE a challenging objective for traditional solvers. We aim to solve this equation for all $x$ and $t$ on a domain $[0, L]$ with periodic boundary conditions $u(0, t) = u(L, t)$ and an initial condition $u(x, 0) = u_0(x)$. The input space is discretized uniformly on a grid of $N_x$ spatial points and $N_t$ time steps. To solve this equation, a neural operator, denoted by NO, is then trained to predict a solution $u(x, t) = u(t)$ given one or multiple previous solutions $u(t - \Delta t)$ with time step $\Delta t$, e.g. $\hat{u}(t) = \text{NO}(u(t - \Delta t))$. Longer trajectory predictions are obtained by feeding the predictions back into the solver, i.e., predicting $u(t + \Delta t)$ from the previous prediction $\hat{u}(t)$ via $\hat{u}(t + \Delta t) = \text{NO}(\hat{u}(t))$. We refer to this process as *unrolling* the model or *rollout*. The goal is to obtain a neural solver that maintains predictions close to the ground truth for as long as possible.

**The MSE training objective**. The most common objective used for training neural solvers is the one-step Mean-Squared Error (MSE) loss: $\mathcal{L}_{\text{MSE}} = \|u(t) - \text{NO}(u(t - \Delta t))\|^2$. By minimizing this one-step MSE, the model learns to replicate the PDE's dynamics, accurately predicting the next step. However, as we roll out the model for long trajectories, the error propagates over time until the predictions start to differ significantly from the ground truth. In Figure 1a the solver is already accurate for 70 seconds, so one might argue that minimizing the one-step MSE is sufficient for achieving long stable rollouts. Yet, the limitations of this approach become apparent when examining the frequency spectrum across the spatial dimension of the ground truth data and resulting predictions. Figure 1b shows that the main dynamics of the KS equation are modeled within a frequency band of low wavenumbers (1 to 25). As a result, the primary errors in predicting a one-step solution arise from inaccuracies in modeling the dynamics of these low frequencies. This is evident in Figure 1c, where the error of the MSE-trained model is smallest for this frequency band relative to the ground truth amplitude. Nonetheless, over a long time horizon, the non-linear term $uu_x$ in the KS equation causes all frequencies to interact, leading to the propagation of high-frequency errors into lower frequencies. Hence, the accurate modeling of frequencies with lower amplitude becomes increasingly important for longer rollout lengths. In the KS equation, this primarily pertains to high frequencies, which the MSE objective significantly neglects.

Based on this analysis, we deduce that in order to obtain long stable rollouts, we need a neural solver that models all spatial frequencies across the spectrum as accurately as possible. Essentially, our objective should give high amplitude frequencies a higher priority, since these are responsible for the main dynamics of the PDE. However, at the same time, the neural solver should not neglect the non-dominant, low amplitude frequency contributions due to their long-term impact on the dynamics.

---

[3]We omit the bold notation for 1D cases where the field $u(x, t)$ is scalar valued.

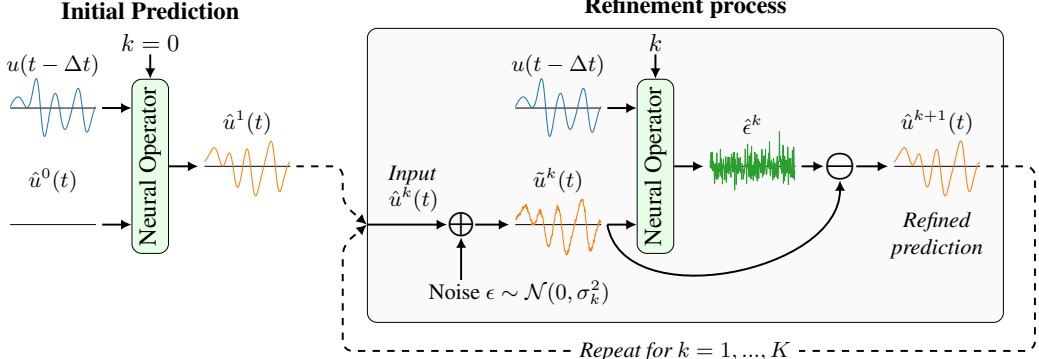

Figure 2: Refinement process of PDE-Refiner during inference. Starting from an initial prediction $\hat{u}^1(t)$, PDE-Refiner uses an iterative refinement process to improve its prediction. Each step represents a denoising process, where the model takes as input the previous step's prediction $u^k(t)$ and tries to reconstruct added noise. By decreasing the noise variance $\sigma_k^2$ over the $K$ refinement steps, PDE-Refiner focuses on all frequencies equally, including low-amplitude information.

## 3 PDE-Refiner

In this section, we present PDE-Refiner, a model that allows for accurate modeling of the solution across all frequencies. The main idea of PDE-Refiner is that we allow the model to look multiple times at its prediction, and, in an iterative manner, improve the prediction. For this, we use a model NO with three inputs: the previous time step(s) $u(t - \Delta t)$, the refinement step index $k \in [0, ..., K]$, and the model's current prediction $\hat{u}^k(t)$. At the first step $k = 0$, we mimic the common MSE objective by setting $\hat{u}^0(t) = 0$ and predicting $u(t)$: $\mathcal{L}^0(u, t) = \|u(t) - \text{NO}\left(\hat{u}^0(t), u(t - \Delta t), 0\right)\|_2^2$. As discussed in Section 2, this prediction will focus on only the dominating frequencies. To improve this prediction, a simple approach would be to train the model to take its own predictions as inputs and output its (normalized) error to the ground truth. However, such a training process has several drawbacks. Firstly, as seen in Figure 1, the dominating frequencies in the data also dominate in the error, thus forcing the model to focus on the same frequencies again. As we empirically verify in Section 4.1, this leads to considerable overfitting and the model does not generalize.

Instead, we propose to implement the refinement process as a denoising objective. At each refinement step $k \geq 1$, we remove low-amplitude information of an earlier prediction by applying noise, e.g. adding Gaussian noise, to the input $\hat{u}^k(t)$ at refinement step $k$: $\tilde{u}^k(t) = \hat{u}^k(t) + \sigma_k \epsilon^k, \epsilon^k \sim \mathcal{N}(0, 1)$. The objective of the model is to predict this noise $\epsilon^k$ and use the prediction $\hat{\epsilon}^k$ to denoise its input: $\hat{u}^{k+1}(t) = \tilde{u}^k(t) - \sigma_k \hat{\epsilon}^k$. By decreasing the noise standard deviation $\sigma_k$ over refinement steps, the model focuses on varying amplitude levels. With the first steps ensuring that high-amplitude information is captured accurately, the later steps focus on low-amplitude information, typically corresponding to the non-dominant frequencies. Generally, we find that an exponential decrease, i.e. $\sigma_k = \sigma_{\min}^{k/K}$ with $\sigma_{\min}$ being the minimum noise standard deviation, works well. The value of $\sigma_{\min}$ is chosen based on the frequency spectrum of the given data. For example, for the KS equation, we use $\sigma_{\min}^2 = 2 \cdot 10^{-7}$. We train the model by denoising ground truth data at different refinement steps:

$$\mathcal{L}^k(u, t) = \mathbb{E}_{\epsilon^k \sim \mathcal{N}(0,1)} \left[\|\epsilon_k - \text{NO}\left(u(t) + \sigma_k \epsilon_k, u(t - \Delta t), k\right)\|_2^2\right] \tag{4}$$

Crucially, by using ground truth samples in the refinement process during training, the model learns to focus on only predicting information with a magnitude below the noise level $\sigma_k$ and ignore potentially larger errors that, during inference, could have occurred in previous steps. To train all refinement steps equally well, we uniformly sample $k$ for each training example: $\mathcal{L}(u, t) = \mathbb{E}_{k \sim U(0,K)}\left[\mathcal{L}^k(u, t)\right]$.

At inference time, we predict a solution $u(t)$ from $u(t - \Delta t)$ by performing the $K$ refinement steps, where we sequentially use the prediction of a refinement step as the input to the next step. While the process allows for any noise distribution, independent Gaussian noise has the preferable property that it is uniform across frequencies. Therefore, it removes information equally for all frequencies, while also creating a prediction target that focuses on all frequencies equally. We empirically verify in Section 4.1 that PDE-Refiner even improves on low frequencies with small amplitudes.

## 3.1 Formulating PDE-Refiner as a Diffusion Model

Denoising processes have been most famously used in diffusion models as well [12, 29–31, 61, 69, 77]. Denoising diffusion probabilistic models (DDPM) randomly sample a noise variable $\mathbf{x}_0 \sim \mathcal{N}(\mathbf{0}, \mathbf{I})$ and sequentially denoise it until the final prediction, $\mathbf{x}_K$, is distributed according to the data:

$$p_\theta(\mathbf{x}_{0:K}) := p(\mathbf{x}_0) \prod_{k=0}^{K-1} p_\theta(\mathbf{x}_{k+1}|\mathbf{x}_k), \quad p_\theta(\mathbf{x}_{k+1}|\mathbf{x}_k) = \mathcal{N}(\mathbf{x}_{k+1}; \boldsymbol{\mu}_\theta(\mathbf{x}_k, k), \boldsymbol{\Sigma}_\theta(\mathbf{x}_k, k)), \quad (5)$$

where $K$ is the number of diffusion steps. For neural PDE solving, one would want $p_\theta(\mathbf{x}_K)$ to model the distribution over solutions, $\mathbf{x}_K = u(t)$, while being conditioned on the previous time step $u(t - \Delta t)$, i.e., $p_\theta(u(t)|u(t - \Delta t))$. For example, Lienen et al. [51] recently proposed DDPMs for modeling 3D turbulent flows due to the flows' unpredictable behavior. Despite the similar use of a denoising process, PDE-Refiner sets itself apart from standard DDPMs in several key aspects. First, diffusion models typically aim to model diverse, multi-modal distributions like in image generation, while the PDE solutions we consider here are deterministic. This necessitates extremely accurate predictions with only minuscule errors. PDE-Refiner accommodates this by employing an exponentially decreasing noise scheduler with a very low minimum noise variance $\sigma_{\min}^2$, decreasing much faster and further than common diffusion schedulers. Second, our goal with PDE-Refiner is not only to model a realistic-looking solution, but also achieve high accuracy across the entire frequency spectrum. Third, we apply PDE-Refiner autoregressively to generate long trajectories. Since neural PDE solvers need to be fast to be an attractive surrogate for classical solvers in applications, PDE-Refiner uses far fewer denoising steps in both training and inferences than typical DDPMs. Lastly, PDE-Refiner directly predicts the signal $u(t)$ at the initial step, while DDPMs usually predict the noise residual throughout the entire process. Interestingly, a similar objective to PDE-Refiner is achieved by the v-prediction [70], which smoothly transitions from predicting the sample $u(t)$ to the additive noise $\epsilon$: $\mathbf{v}^k = \sqrt{1 - \sigma_k^2}\epsilon - \sigma_k u(t)$. Here, the first step $k = 0$, yields the common MSE prediction objective by setting $\sigma_0 = 1$. With an exponential noise scheduler, the noise variance is commonly much smaller than 1 for $k \geq 1$. In these cases, the weight of the noise is almost 1 in the v-prediction, giving a diffusion process that closely resembles PDE-Refiner.

Nonetheless, the similarities between PDE-Refiner and DDPMs indicate that PDE-Refiner has a potential interpretation as a probabilistic latent variable model. Thus, by sampling different noises during the refinement process, PDE-Refiner may provide well-calibrated uncertainties which faithfully indicate when the model might be making errors. We return to this intriguing possibility later in Section 4.1. Further, we find empirically that implementing PDE-Refiner as a diffusion model with our outlined changes in the previous paragraph, versus implementing it as an explicit denoising process, obtains similar results. The benefit of implementing PDE-Refiner as a diffusion model is the large literature on architecture and hyperparameter studies, as well as available software for diffusion models. Hence, we use a diffusion-based implementation of PDE-Refiner in our experiments.

## 4 Experiments

We demonstrate the effectiveness of PDE-Refiner on a diverse set of common PDE benchmarks. In 1D, we study the Kuramoto-Sivashinsky equation and compare to typical temporal rollout methods. Further, we study the models' robustness to different spatial frequency spectra by varying the visocisity in the KS equation. In 2D, we compare PDE-Refiner to hybrid PDE solvers on a turbulent Kolmogorov flow, and provide a speed comparison between solvers. We make our code publicly available at https://github.com/microsoft/pdearena.

### 4.1 Kuramoto-Sivashinsky 1D equation

**Experimental setup**. We evaluate PDE-Refiner and various baselines on the Kuramoto-Sivashinsky 1D equation. We follow the data generation setup of Brandstetter et al. [8] by using a mesh of length $L$ discretized uniformly for 256 points with periodic boundaries. For each trajectory, we randomly sample the length $L$ between $[0.9 \cdot 64, 1.1 \cdot 64]$ and the time step $\Delta t \sim U(0.18, 0.22)$. The initial conditions are sampled from a distribution over truncated Fourier series with random coefficients $\{A_m, \ell_m, \phi_m\}_m$ as $u_0(x) = \sum_{m=1}^{10} A_m \sin(2\pi \ell_m x / L + \phi_m)$. We generate a training dataset with 2048 trajectories of rollout length $140\Delta t$, and test on 128 trajectories with a duration

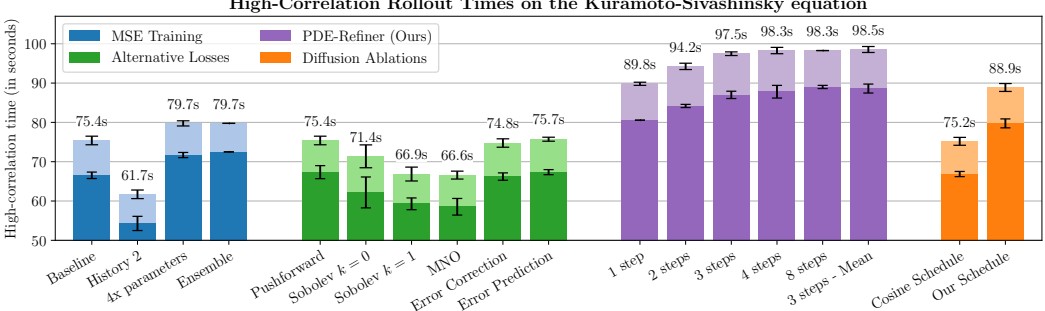

Figure 3: Experimental results on the Kuramoto-Sivashinsky equation. Dark and light colors indicate time for average correlation to drop below 0.9 and 0.8, respectively. Error bars represent standard deviation for 5 seeds. We distinguish four model groups: models trained with the common one-step MSE (left), alternative losses considered in previous work (center left), our proposed PDE-Refiner (center right), and denoising diffusion (center right). All models use a modern U-Net neural operator [22]. PDE-Refiner surpasses all baselines with accurate rollouts up to nearly 100 seconds.

of $640\Delta t$. As the network architecture, we use the modern U-Net of Gupta et al. [22] with hidden size 64 and 3 downsampling layers. U-Nets have demonstrated strong performance in both neural PDE solving [22, 56, 80] and diffusion modeling [29, 30, 61], making it an ideal candidate for PDE-Refiner. A common alternative is the Fourier Neural Operator (FNO) [49]. Since FNO layers cut away high frequencies, we find them to perform suboptimally on predicting the residual noise in PDE-Refiner and DDPMs. Yet, our detailed study with FNOs in Appendix E.1 shows that even here PDE-Refiner offers significant performance gains. Finally, we also evaluate on Dilated ResNets [78] in Appendix E.2, showing very similar results to the U-Net. Since neural surrogates can operate on larger time steps, we directly predict the solution at every 4th time step. In other words, to predict $u(t)$, each model takes as input the previous time step $u(t - 4\Delta t)$ and the trajectory parameters $L$ and $\Delta t$. Thereby, the models predict the residual between time steps $\Delta u(t) = u(t) - u(t - 4\Delta t)$ instead of $u(t)$ directly, which has shown superior performance at this timescale [50]. Ablations on the time step size can be found in Appendix E.3. As evaluation criteria, we report the model rollouts' high-correlation time [43, 79]. For this, we autoregressively rollout the models on the test set and measure the Pearson correlation between the ground truth and the prediction. We then report the time when the average correlation drops below 0.8 and 0.9, respectively, to quantify the time horizon for which the predicted rollouts remain accurate. We investigate other evaluation criteria such as mean-squared error and varying threshold values in Appendix D, leading to the same conclusions.

**MSE Training**. We compare PDE-Refiner to three groups of baselines in Figure 3. The first group are models trained with the one-step MSE error, i.e., predicting $\Delta u(t)$ from $u(t - 4\Delta t)$. The baseline U-Net obtains a high-correlation rollout time of 75 seconds, which corresponds to 94 autoregressive steps. We find that incorporating more history information as input, i.e. $u(t - 4\Delta t)$ and $u(t - 8\Delta t)$, improves the one-step prediction but worsens rollout performance. The problem arising is that the difference between the inputs $u(t - 4\Delta t) - u(t - 8\Delta t)$ is highly correlated with the model's target $\Delta u(t)$, the residual of the next time step. This leads the neural operator to focus on modeling the second-order difference $\Delta u(t) - \Delta u(t - 4\Delta t)$. As observed in classical solvers [36], using higher-order differences within an explicit autoregressive scheme is known to deteriorate the rollout stability and introduce exponentially increasing errors over time. We include further analysis of this behavior in Appendix E.4. Finally, we verify that PDE-Refiner's benefit is not just because of having an increased model complexity by training a model with 4 times the parameter count and observe a performance increase performance by only 5%. Similarly, averaging the predictions of an ensemble of 5 MSE-trained models cannot exceed 80 seconds of accurate rollouts.

**Alternative losses**. The second baseline group includes alternative losses and post-processing steps proposed by previous work to improve rollout stability. The pushforward trick [9] rolls out the model during training and randomly replaces ground truth inputs with model predictions. This trick does not improve performance in our setting, confirming previous results [8]. While addressing potential input distribution shift, the pushforward trick cannot learn to include the low-amplitude information for accurate long-term predictions, as no gradients are backpropagated through the predicted input for stability reasons. Focusing more on high-frequency information, the Sobolev norm loss [50] maps the

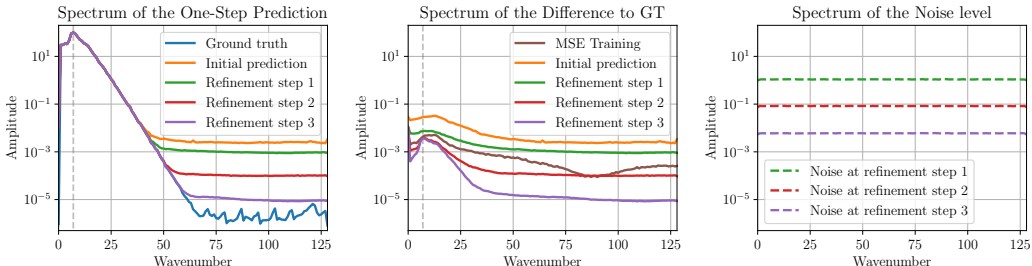

Figure 4: Analyzing the prediction errors of PDE-Refiner and the MSE training in frequency space over the spatial dimension. **Left**: the spectrum of intermediate predictions $\hat{u}^0(t), \hat{u}^1(t), \hat{u}^2(t), \hat{u}^3(t)$ of PDE-Refiner's refinement process compared to the Ground Truth. **Center**: the spectrum of the difference between ground truth and intermediate predictions, i.e. $|\text{FFT}(u(t) - \hat{u}^k(t))|$. **Right**: the spectrum of the noise $\sigma_k \epsilon^k$ added at different steps of the refinement process. Any error with lower amplitude will be significantly augmented during denoising.

prediction error into the frequency domain and weighs all frequencies equally for $k = 0$ and higher frequencies more for $k = 1$. However, focusing on high-frequency information leads to a decreased one-step prediction accuracy for the high-amplitude frequencies, such that the rollout time shortens. The Markov Neural Operator (MNO) [50] additionally encourages dissipativity via regularization, but does not improve over the common Sobolev norm losses. Inspired by McGreivy et al. [58], we report the rollout time when we correct the predictions of the MSE models for known invariances in the equation. We ensure mass conservation by zeroing the mean and set any frequency above 60 to 0, as their amplitude is below `float32` precision (see Appendix D.1). This does not improve over the original MSE baselines, showing that the problem is not just an overestimate of the high frequencies, but the accurate modeling of a broader spectrum of frequencies. Finally, to highlight the advantages of the denoising process in PDE-Refiner, we train a second model to predict another MSE-trained model's errors (Error Prediction). This model quickly overfits on the training dataset and cannot provide gains for unseen trajectories, since it again focuses on the same high-amplitude frequencies.

**PDE-Refiner - Number of refinement steps**.    Figure 3 shows that PDE-Refiner significantly outperforms the baselines and reaches almost 100 seconds of stable rollout. Thereby, we have a trade-off between number of refinement steps and performance. When training PDE-Refiner with 1 to 8 refinement steps, we see that the performance improves with more refinement steps, but more steps require more model calls and thus slows down the solver. However, already using a single refinement step improves the rollout performance by 20% over the best baseline, and the gains start to flatten at 3 to 4 steps. Thus, for the remainder of the analysis, we will focus on using 3 refinement steps.

**Diffusion Ablations**.    In an ablation study of PDE-Refiner, we evaluate a standard denoising diffusion model [30] that we condition on the previous time step $u(t - 4\Delta t)$. When using a common cosine noise schedule [61], the model performs similar to the MSE baselines. However, with our exponential noise decrease and lower minimum noise level, the diffusion models improve by more than 10 seconds. Using the prediction objective of PDE-Refiner gains yet another performance improvement while reducing the number of sampling steps significantly. Furthermore, to investigate the probabilistic nature of PDE-Refiner, we check whether it samples single modes under potentially multi-modal uncertainty. For this, we average 16 samples at each rollout time step (`3 steps - Mean` in Figure 3) and find slight performance improvements, indicating that PDE-Refiner mostly predicts single modes.

**Modeling the Frequency Spectrum**.    We analyse the performance difference between the MSE training and PDE-Refiner by comparing their one-step prediction in the frequency domain in Figure 4. Similar to the MSE training, the initial prediction of PDE-Refiner has a close-to uniform error pattern across frequencies. While the first refinement step shows an improvement across all frequencies, refinement steps 2 and 3 focus on the low-amplitude frequencies and ignore higher amplitude errors. This can be seen by the error for wavenumber 7, i.e., the frequency with the highest input amplitude, not improving beyond the first refinement step. Moreover, the MSE training obtains almost the identical error rate for this frequency, emphasizing the importance of low-amplitude information. For all other frequencies, PDE-Refiner obtains a much lower loss, showing its improved accuracy on low-amplitude information over the MSE training. We highlight that PDE-Refiner does not only improve the high frequencies, but also the lowest frequencies (wavenumber 1-6) with low amplitude.

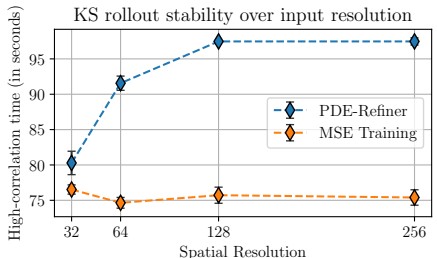
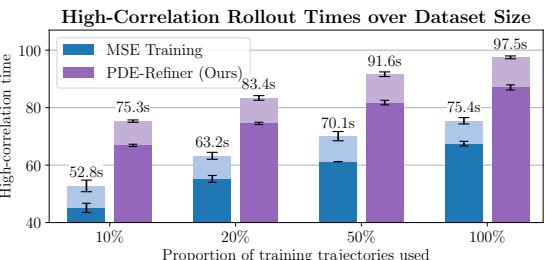

Figure 5: **Left**: Stable rollout time over input resolution. PDE-Refiner models the high frequencies to improve its rollout on higher resolutions. **Right**: Training PDE-Refiner and the MSE baseline on smaller datasets. PDE-Refiner consistently outperforms the MSE baseline, increasing its relative improvement to 50% for the lowest data regime.

**Input Resolution.** We demonstrate that capturing high-frequency information is crucial for PDE-Refiner's performance gains over the MSE baselines by training both models on datasets of subsampled spatial resolution. With lower resolution, fewer frequencies are present in the data and can be modeled. As seen in Figure 5, MSE models achieve similar rollout times for resolutions between 32 and 256, emphasizing its inability to model high-frequency information. At a resolution of 32, PDE-Refiner achieves similar performance to the MSE baseline due to the missing high-frequency information. However, as resolution increases, PDE-Refiner significantly outperforms the baseline, showcasing its utilization of high-frequency information.

**Spectral data augmentation.** A pleasant side effect of PDE-Refiner is data augmentation, which is induced by adding varying Gaussian noise $\sigma_k \epsilon^k, \epsilon^k \sim \mathcal{N}(0, 1)$ at different stages $k$ of the refinement process. Effectively, data augmentation is achieved by randomly distorting the input at different scales, and forcing the model to recover the underlying structure. This gives an ever-changing input and objective, forces the model to fit different parts of the spectrum, and thus making it more difficult for the model to overfit. Compared to previous works such as Lie Point Symmetry data augmentation [8] or general covariance and random coordinate transformations [15], the data augmentation in PDE-Refiner is purely achieved by adding noise, and thus very simple and applicable to any PDE. While we leave more rigorous testing of PDE-Refiner induced data augmentation for future work, we show results for the low training data regime in Figure 5. When training PDE-Refiner and the MSE baseline on 10%, 20% and 50% of the training trajectories, PDE-Refiner consistently outperforms the baseline in all settings. Moreover, with only 10% of the training data, PDE-Refiner performs on par to the MSE model at 100%. Finally, the relative improvement of PDE-Refiner increases to 50% for this low data regime, showing its objective acting as data augmentation and benefiting generalization.

**Uncertainty estimation.** When applying neural PDE solvers in practice, knowing how long the predicted trajectories remain accurate is crucial. To estimate PDE-Refiner's predictive uncertainty, we sample 32 rollouts for each test trajectory by generating different Gaussian noise during the refinement process. We compute the time when the samples diverge from one another, i.e. their cross-correlation goes below 0.8, and investigate whether this can be used to accurately estimate how long the model's rollouts remain close to the ground truth. Figure 6 shows that the cross-correlation time between samples closely aligns with the time over which the rollout remains accurate, leading to a $R^2$ coefficient of 0.86 between the two times. Furthermore, the prediction for how long the rollout remains accurate depends strongly on the individual trajectory – PDE-Refiner reliably identifies trajectories that are easy or challenging to roll out from. In Appendix E.5, we compare PDE-Refiner's uncertainty estimate to two other

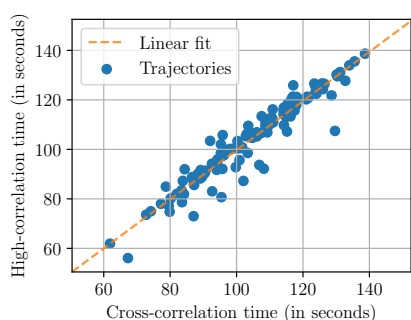

Figure 6: Uncertainty estimate of PDE-Refiner. Each point represents the estimated correlation time via sample cross-correlation (x-axis) and the ground truth time (y-axis) for a test trajectory.

common approaches. PDE-Refiner provides more accurate estimates than input modulation [5, 71], while only requiring one trained model compared to a model ensemble [45, 71].

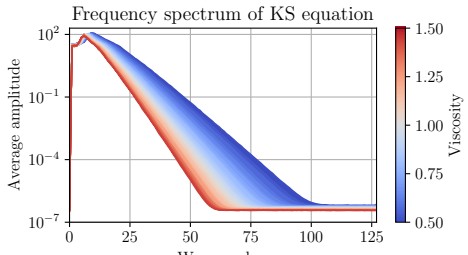
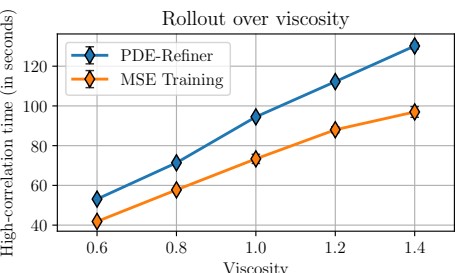

Figure 7: Visualizing the parameter-dependent KS equation. **Left**: Frequency spectrum of ground truth trajectories over viscosities. **Right**: Accurate rollout time over viscosities (error bars neglected if smaller than marker size). PDE-Refiner obtains improvements across all viscosities.

## 4.2 Parameter-dependent KS equation

So far, we have focused on the KS equation with a viscosity term of $\nu = 1$. Under varying values of $\nu$, the Kuramoto-Sivashinsky equation has been shown to develop diverse behaviors and fixed points [35, 40, 74]. This offers an ideal benchmark for evaluating neural surrogate methods on a diverse set of frequency spectra. We generate 4096 training and 512 test trajectories with the same data generation process as before, except that for each trajectory, we sample $\nu$ uniformly between 0.5 and 1.5. This results in the spatial frequency spectrum of Figure 7, where high frequencies are damped for larger viscosities but amplified for lower viscosities. Thus, an optimal neural PDE solver for this dataset needs to work well across a variety of frequency spectra. We keep the remaining experimental setup identical to Section 4.1, and add the viscosity $\nu$ to the conditioning set of the neural operators.

We compare PDE-Refiner to an MSE-trained model by plotting the stable rollout time over viscosities in Figure 7. Each marker represents between for trajectories in $[\nu - 0.1, \nu + 0.1]$. PDE-Refiner is able to get a consistent significant improvement over the MSE model across viscosities, verifying that PDE-Refiner works across various frequency spectra and adapts to the given underlying data. Furthermore, both models achieve similar performance to their unconditional counterpart for $\nu = 1.0$. This again highlights the strength of the U-Net architecture and baselines we consider here.

## 4.3 Kolmogorov 2D Flow

**Simulated data**. As another common fluid-dynamics benchmark, we apply PDE-Refiner to the 2D Kolmogorov flow, a variant of the incompressible Navier-Stokes flow. The PDE is defined as:

$$\partial_t \mathbf{u} + \nabla \cdot (\mathbf{u} \otimes \mathbf{u}) = \nu \nabla^2 \mathbf{u} - \frac{1}{\rho} \nabla p + \mathbf{f} \tag{6}$$

where $\mathbf{u} : [0, T] \times \mathcal{X} \to \mathbb{R}^2$ is the solution, $\otimes$ the tensor product, $\nu$ the kinematic viscosity, $\rho$ the fluid density, $p$ the pressure field, and, finally, $\mathbf{f}$ the external forcing. Following previous work [43, 79], we set the forcing to $\mathbf{f} = \sin(4y)\hat{\mathbf{x}} - 0.1\mathbf{u}$, the density $\rho = 1$, and viscosity $\nu = 0.001$, which corresponds to a Reynolds number of 1000. The ground truth data is generated using a finite volume-based direct numerical simulation (DNS) method [43, 57] with a time step of $\Delta t = 7.0125 \times 10^{-3}$ and resolution of $2048 \times 2048$, and afterward downscaled to $64 \times 64$. To align our experiments with previous results, we use the same dataset of 128 trajectories for training and 16 trajectories for testing as Sun et al. [79].

**Experimental setup**. We employ a modern U-Net [22] as the neural operator backbone. Due to the lower input resolution, we set $\sigma_{\min}^2 = 10^{-3}$ and use 3 refinement steps in PDE-Refiner. For efficiency, we predict 16 steps ($16\Delta t$) into the future and use the difference $\Delta \mathbf{u} = \mathbf{u}(t) - \mathbf{u}(t - 16\Delta t)$ as the output target. Besides the MSE objective, we compare PDE-Refiner with FNOs [50], classical PDE solvers (i.e., DNS) on different resolutions, and state-of-the-art hybrid machine learning solvers [43, 79], which estimate the convective flux $\mathbf{u} \otimes \mathbf{u}$ via neural networks. Learned Interpolation (LI) [43] takes the previous solution $\mathbf{u}(t - \Delta t)$ as input to predict $\mathbf{u}(t)$, similar to PDE-Refiner. In contrast, the Temporal Stencil Method (TSM) Sun et al. [79] combines information from multiple previous time steps using HiPPO features [19, 20]. We also compare PDE-Refiner to a Learned Correction model (LC) [43, 83], which corrects the outputs of a classical solver with neural networks. For evaluation, we roll out the models on the 16 test trajectories and determine the Pearson correlation with the

ground truth in terms of the scalar vorticity field $\omega = \partial_x u_y - \partial_y u_x$. Following previous work [79], we report in Table 1 the time until which the average correlation across trajectories falls below 0.8.

**Results.** Similar to previous work [22, 55], we find that modern U-Nets outperform FNOs on the 2D domain for long rollouts. Our MSE-trained U-Net already surpasses all classical and hybrid PDE solvers. This result highlights the strength of our baselines, and improving upon those poses a significant challenge. Nonetheless, PDE-Refiner manages to provide a substantial gain in performance, remaining accurate 32% longer than the best single-input hybrid method and 10% longer than the best multi-input hybrid methods and MSE model. We reproduce the frequency plots of Figure 4 for this dataset in Appendix E.6. The plots exhibit a similar behavior of both models. Compared to the KS equation, the Kolmogorov flow has a shorter (due to the resolution) and flatter spatial frequency spectrum. This accounts for the smaller relative gain of PDE-Refiner on the MSE baseline here.

Table 1: Duration of high correlation ($>$ 0.8) on the 2D Kolmogorov flow. Results for classical PDE solvers and hybrid methods taken from Sun et al. [79].

| Method | Corr. $> 0.8$ time |
|---|---|
| *Classical PDE Solvers* | |
| DNS - $64 \times 64$ | 2.805 |
| DNS - $128 \times 128$ | 3.983 |
| DNS - $256 \times 256$ | 5.386 |
| DNS - $512 \times 512$ | 6.788 |
| DNS - $1024 \times 1024$ | 8.752 |
| *Hybrid Methods* | |
| LC [43, 83] - CNN | 6.900 |
| LC [43, 83] - FNO | 7.630 |
| LI [43] - CNN | 7.910 |
| TSM [79] - FNO | 7.798 |
| TSM [79] - CNN | 8.359 |
| TSM [79] - HiPPO | 9.481 |
| *ML Surrogates* | |
| MSE training - FNO | $6.451 \pm 0.105$ |
| MSE training - U-Net | $9.663 \pm 0.117$ |
| PDE-Refiner - U-Net | $\mathbf{10.659} \pm 0.092$ |

**Speed comparison.** We evaluate the speed of the rollout generation for the test set (16 trajectories of 20 seconds) of three best solvers on an NVIDIA A100 GPU. The MSE U-Net generates the trajectories in 4.04 seconds ($\pm 0.01$), with PDE-Refiner taking 4 times longer ($16.53 \pm 0.04$ seconds) due to four model calls per step. With that, PDE-Refiner is still faster than the best hybrid solver, TSM, which needs 20.25 seconds ($\pm 0.05$). In comparison to the ground truth solver at resolution $2048 \times 2048$ with 31 minute generation time on GPU, all surrogates provide a significant speedup.

## 5   Conclusion

In this paper, we conduct a large-scale analysis of temporal rollout strategies for neural PDE solvers, identifying that the neglect of low-amplitude information often limits accurate rollout times. To address this issue, we introduce PDE-Refiner, which employs an iterative refinement process to accurately model all frequency components. This approach remains considerably longer accurate during rollouts on three fluid dynamic datasets, effectively overcoming the common pitfall.

**Limitations.** The primary limitation of PDE-Refiner is its increased computation time per prediction. Although still faster than hybrid and classical solvers, future work could investigate reducing compute for early refinement steps, or applying distillation and enhanced samplers to accelerate refinement, as seen in diffusion models [3, 38, 70, 88]. Another challenge is the smaller gain of PDE-Refiner with FNOs due to the modeling of high-frequency noise, which thus presents an interesting avenue for future work. Further architectures like Transformers [13, 84] can be explored too, having been shown to also suffer from spatial frequency biases for PDEs [11]. Additionally, most of our study focused on datasets where the test trajectories come from a similar domain as the training. Evaluating the effects on rollout in inter- and extrapolation regimes, e.g. on the viscosity of the KS dataset, is left for future work. Lastly, we have only investigated additive Gaussian noise. Recent blurring diffusion models [32, 47] focus on different spatial frequencies over the sampling process, making them a potentially suitable option for PDE solving as well.

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

# SUPPLEMENTARY MATERIAL
# PDE-REFINER: ACHIEVING ACCURATE LONG ROLLOUTS WITH NEURAL PDE SOLVERS

## TABLE OF CONTENTS

## A   Broader Impact

Neural PDE solvers hold significant potential for offering computationally cheaper approaches to modeling a wide range of natural phenomena than classical solvers. As a result, PDE surrogates could potentially contribute to advancements in various research fields, particularly within the natural sciences, such as fluid dynamics and weather modeling. Further, reducing the compute needed for simulations may reduce the carbon footprint of research institutes and industries that rely on such models. Our proposed method, PDE-Refiner, can thereby help in improving the accuracy of these neural solvers, particularly for long-horizon predictions, making their application more viable.

However, it is crucial to note that reliance on simulations necessitates rigorous cross-checks and continuous monitoring. This is particularly true for neural surrogates, which may have been trained on simulations themselves and could introduce additional errors when applied to data outside its original training distribution. Hence, it is crucial for the underlying assumptions and limitations of these surrogates to be well-understood in applications.

## B   Reproducibility Statement

To ensure reproducibility, we publish our code at `https://github.com/microsoft/pdearena`. We report the used model architectures, hyperparameters, and dataset properties in detail in Section 4 and Appendix D. We additionally include pseudocode for our proposed method, PDE-Refiner, in Appendix C. All experiments on the KS datasets have been repeated for five seeds, and three seeds have been used for the Kolmogorov Flow dataset. Plots and tables with quantitative results show the standard deviation across these seeds.

As existing software assets, we base our implementation on the PDE-Arena [22], which implements a Python-based training framework for neural PDE solvers in PyTorch [62] and PyTorch Lightning [14]. For the diffusion models, we use the library diffusers [65]. We use Matplotlib [34] for plotting and NumPy [89] for data handling. For data generation, we use scipy [85] in the public code of Brandstetter et al. [8] for the KS equation, and JAX [6] in the public code of Kochkov et al. [43], Sun et al. [79] for the 2D Kolmogorov Flow dataset. The usage of these assets is further described in Appendix D.

In terms of computational resources, all experiments have been performed on NVIDIA V100 GPUs with 16GB memory. For the experiments on the KS equation, each model was trained on a single NVIDIA V100 for 1 to 2 days. We note that since the model errors are becoming close to the float32 precision limit, the results may differ on other GPU architectures (e.g. A100), especially when different precision like tensorfloat (TF32) or float16 are used in the matrix multiplications. This can artificially limit the possible accurate rollout time a model can achieve. For reproducibility, we recommend using V100 GPUs. For the 2D Kolmogorov Flow dataset, we parallelized the models across 4 GPUs, with a training time of 2 days. The speed comparison for the 2D Kolmogorov Flow were performed on an NVIDIA A100 GPU with 80GB memory. Overall, the experiments in this paper required roughly 250 GPU days, with additional 400 GPU days for development, hyperparameter search, and the supplementary results in Appendix E.

# C   PDE-Refiner - Pseudocode

In this section, we provide pseudocode to implement PDE-Refiner in Python with common deep learning frameworks like PyTorch [62] and JAX [6]. The hyperparameters to PDE-Refiner are the number of refinement steps $K$, called `num_steps` in the pseudocode, and the minimum noise standard deviation $\sigma_{\min}$, called `min_noise_std`. Further, the neural operator `NO` can be an arbitrary network architecture, such as a U-Net as in our experiments, and is represented by `MyNetwork` / `self.neural_operator` in the code.

The dynamics of PDE-Refiner can be implemented via three short functions. The `train_step` function takes as input a training example of solution $u(t)$ (named `u_t`) and the previous solution $u(t - \Delta t)$ (named `u_prev`). We uniformly sample the refinement step we want to train, and use the classical MSE objective if $k = 0$. Otherwise, we train the model to denoise $u(t)$. The `loss` can be used to calculate gradients and update the parameters with common optimizers. The operation `randn_like` samples Gaussian noise of the same shape as `u_t`. Further, for batch-wise inputs, we sample $k$ for each batch element independently. For inference, we implement the function `predict_next_solution`, which iterates through the refinement process of PDE-Refiner. Lastly, to generate a trajectory from an initial condition `u_initial`, the function `rollout` autoregressively predicts the next solutions. This gives us the following pseudocode:

```python
class PDERefiner:
    def __init__(self, num_steps, min_noise_std):
        self.num_steps = num_steps
        self.min_noise_std = min_noise_std
        self.neural_operator = MyNetwork(...)

    def train_step(self, u_t, u_prev):
        k = randint(0, self.num_steps + 1)
        if k == 0:
            pred = self.neural_operator(zeros_like(u_t), u_prev, k)
            target = u_t
        else:
            noise_std = self.min_noise_std ** (k / self.num_steps)
            noise = randn_like(u_t)
            u_t_noised = u_t + noise * noise_std
            pred = self.neural_operator(u_t_noised, u_prev, k)
            target = noise
        loss = mse(pred, target)
        return loss

    def predict_next_solution(self, u_prev):
        u_hat_t = self.neural_operator(zeros_like(u_prev), u_prev, 0)
        for k in range(1, self.num_steps + 1):
            noise_std = self.min_noise_std ** (k / self.num_steps)
            noise = randn_like(u_t)
            u_hat_t_noised = u_hat_t + noise * noise_std
            pred = self.neural_operator(u_hat_t_noised, u_prev, k)
            u_hat_t = u_hat_t_noised - pred * noise_std
        return u_hat_t

    def rollout(self, u_initial, timesteps):
        trajectory = [u_initial]
        for t in range(timesteps):
            u_hat_t = self.predict_next_solution(trajectory[-1])
            trajectory.append(u_hat_t)
        return trajectory
```

As discussed in Section 3.1, PDE-Refiner can be alternatively implemented as a diffusion model. To demonstrate this implementation, we use the Python library diffusers [65] (version 0.15) in the pseudocode below. We create a DDPM scheduler where we set the number of diffusion steps to the number of refinement steps and the prediction type to `v_prediction` [70]. Further, for simplicity,

we set the betas to the noise variances of PDE-Refiner. We note that in diffusion models and in diffusers, the noise variance $\sigma_k^2$ at diffusion step $k$ is calculated as:

$$\sigma_k^2 = 1 - \bar{\alpha}_k = 1 - \prod_{\kappa=k}^{K}(1 - \beta_\kappa) = 1 - \prod_{\kappa=k}^{K}(1 - \sigma_{\min}^{2\kappa/K})$$

Since we generally use few diffusion steps such that the noise variance falls quickly, i.e. $\sigma_{\min}^{2k/K} \gg \sigma_{\min}^{2(k+1)/K}$, the product in above's equation is dominated by the last term $1 - \sigma_{\min}^{2k/K}$. Thus, the noise variances in diffusion are $\sigma_k^2 \approx \sigma_{\min}^{2k/K}$. Further, for $k = 0$ and $k = K$, the two variances are always the same since the product is 0 or a single element, respectively. If needed, one could correct for the product terms in the intermediate variances. However, as we show in Appendix E.7, PDE-Refiner is robust to small changes in the noise variance and no performance difference was notable. With this in mind, PDE-Refiner can be implemented as follows:

```python
from diffusers.schedulers import DDPMScheduler

class PDERefinerDiffusion:
    def __init__(self, num_steps, min_noise_std):
        betas = [min_noise_std ** (k / num_steps)
                    for k in reversed(range(num_steps + 1))]
        self.scheduler = DDPMScheduler(num_train_timesteps=num_steps + 1,
                                        trained_betas=betas,
                                        prediction_type='v_prediction',
                                        clip_sample=False)
        self.num_steps = num_steps
        self.neural_operator = MyNetwork(...)

    def train_step(self, u_t, u_prev):
        k = randint(0, self.num_steps + 1)
        # The scheduler uses t=K for first step prediction, and t=0 for minimum noise.
        # To be consistent with the presentation in the paper, we keep k and the
        # scheduler time separate. However, one can also use the scheduler time step
        # as k directly and acts as conditional input to the neural operator.
        scheduler_t = self.num_steps - k
        noise_factor = self.scheduler.alphas_cumprod[scheduler_t]
        signal_factor = 1 - noise_factor
        noise = randn_like(u_t)
        u_t_noised = self.scheduler.add_noise(u_t, noise, scheduler_t)
        pred = self.neural_operator(u_t_noised, u_prev, k)
        target = (noise_factor ** 0.5) * noise - (signal_factor ** 0.5) * u_t
        loss = mse(pred, target)
        return loss

    def predict_next_solution(self, u_prev):
        u_hat_t_noised = randn_like(u_prev)
        for scheduler_t in self.scheduler.timesteps:
            k = self.num_steps - scheduler_t
            pred = self.neural_operator(u_hat_t_noised, u_prev, k)
            out = self.scheduler.step(pred, scheduler_t, u_hat_t_noised)
            u_hat_t_noised = out.prev_sample
        u_hat_t = u_hat_t_noised
        return u_hat_t

    def rollout(self, u_initial, timesteps):
        trajectory = [u_initial]
        for t in range(timesteps):
            u_hat_t = self.predict_next_solution(trajectory[-1])
            trajectory.append(u_hat_t)
        return trajectory
```

**Training examples**

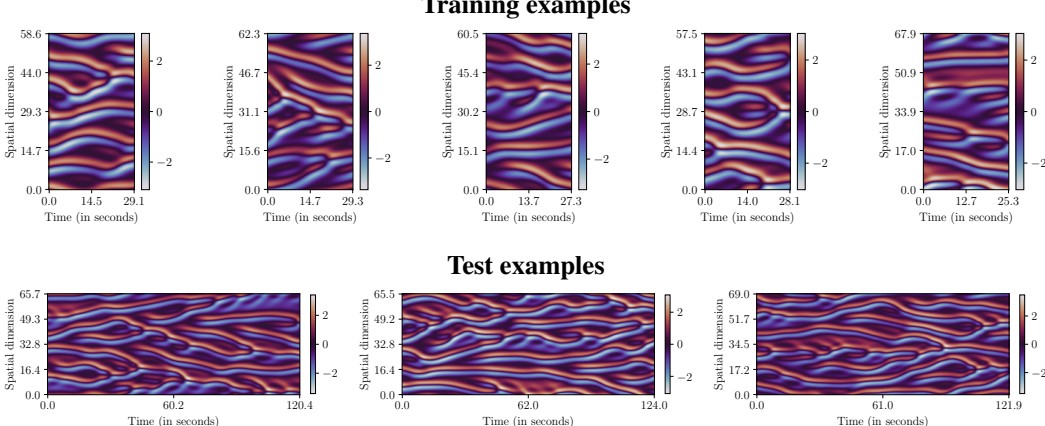

Figure 8: Dataset examples of the Kuramoto-Sivashinsky dataset. The training trajectories are generated with 140 time steps, while the test trajectories consist of 640 time steps. The spatial dimension is uniformly sampled from $[0.9 \cdot 64, 1.1 \cdot 64]$, and the time step in seconds from $[0.18, 0.22]$.

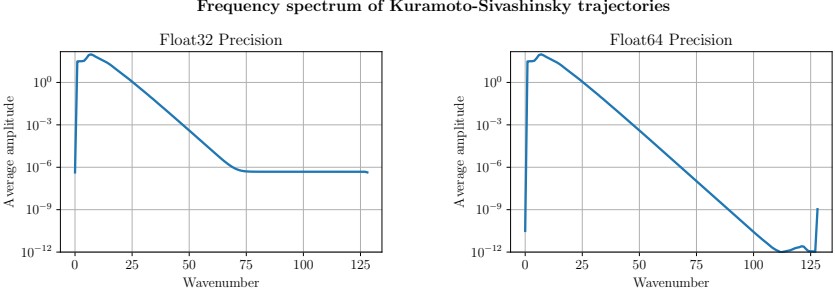

Figure 9: Frequency spectrum of the Kuramoto-Sivashinsky dataset under different precisions. Casting the input data to `float32` precision removes the high frequency information due to adding noise with higher amplitude. Neural surrogates trained on `float64` did not improve over `float32`, showing that it does not affect models in practice.

## D Experimental details

In this section, we provide a detailed description of the data generation, model architecture, and hyper-parameters used in our three datasets: Kuramoto-Sivashinsky (KS) equation, parameter-dependent KS equation, and the 2D Kolmogorov flow. Additionally, we provide an overview of all results with corresponding error bars in numerical table form. Lastly, we show example trajectories for each dataset.

### D.1 Kuramoto-Sivashinsky 1D dataset

**Data generation**. We follow the data generation setup of Brandstetter et al. [8], which uses the method of lines with the spatial derivatives computed using the pseudo-spectral method. For each trajectory in our dataset, the first 360 solution steps are truncated and considered as a warmup for the solver. For further details on the data generation setup, we refer to Brandstetter et al. [8].

Our dataset can be reproduced with the public code[4] of Brandstetter et al. [8]. To obtain the training data, the data generation command in the repository needs to be adjusted by setting the number of training samples to 2048, and 0 for both validation and testing. For validation and testing, we increase the rollout time by adding the arguments `--nt=1000 --nt_effective=640 --end_time=200`, and setting the number of samples to 128 each. We provide training and test examples in Figure 8.

---

[4]`https://github.com/brandstetter-johannes/LPSDA#produce-datasets-for-kuramoto-shivashinsky-ks-equation`

Table 2: Detailed list of layers in the deployed modern U-Net. The parameter *channels* next to a layer represents the number of feature channels of the layer's output. The U-Net uses the four different channel sizes $c_1, c_2, c_3, c_4$, which are hyperparameters. The skip connection from earlier layers in a residual block is implemented by concatenating the features before the first GroupNorm. For the specifics of the residual blocks, see Figure 10.

| Index | Layer |
|-------|-------|
| | *Encoder* |
| 1 | Conv(kernel size=3, channels=$c_1$, stride=1) |
| 2 | ResidualBlock(channels=$c_1$) |
| 3 | ResidualBlock(channels=$c_1$) |
| 4 | Conv(kernel size=3, channels=$c_1$, stride=2) |
| 5 | ResidualBlock(channels=$c_2$) |
| 6 | ResidualBlock(channels=$c_2$) |
| 7 | Conv(kernel size=3, channels=$c_2$, stride=2) |
| 8 | ResidualBlock(channels=$c_3$) |
| 9 | ResidualBlock(channels=$c_3$) |
| 10 | Conv(kernel size=3, channels=$c_3$, stride=2) |
| 11 | ResidualBlock(channels=$c_4$) |
| 12 | ResidualBlock(channels=$c_4$) |
| | *Middle block* |
| 13 | ResidualBlock(channels=$c_4$) |
| 14 | ResidualBlock(channels=$c_4$) |
| | *Decoder* |
| 15 | ResidualBlock(channels=$c_4$, skip connection from Layer 12) |
| 16 | ResidualBlock(channels=$c_4$, skip connection from Layer 11) |
| 17 | ResidualBlock(channels=$c_3$, skip connection from Layer 10) |
| 18 | TransposeConvolution(kernel size=4, channels=$c_3$, stride=2) |
| 19 | ResidualBlock(channels=$c_3$, skip connection from Layer 9) |
| 20 | ResidualBlock(channels=$c_3$, skip connection from Layer 8) |
| 21 | ResidualBlock(channels=$c_2$, skip connection from Layer 7) |
| 22 | TransposeConvolution(kernel size=4, channels=$c_3$, stride=2) |
| 19 | ResidualBlock(channels=$c_2$, skip connection from Layer 6) |
| 20 | ResidualBlock(channels=$c_2$, skip connection from Layer 5) |
| 21 | ResidualBlock(channels=$c_1$, skip connection from Layer 4) |
| 22 | TransposeConvolution(kernel size=4, channels=$c_3$, stride=2) |
| 23 | ResidualBlock(channels=$c_1$, skip connection from Layer 3) |
| 24 | ResidualBlock(channels=$c_1$, skip connection from Layer 2) |
| 25 | ResidualBlock(channels=$c_1$, skip connection from Layer 1) |
| 26 | GroupNorm(channels=$c_1$, groups=8) |
| 27 | GELU activation |
| 28 | Convolution(kernel size=3, channels=1, stride=1) |

The data is generated with `float64` precision, and afterward converted to `float32` precision for storing and training of the neural surrogates. Since we convert the precision in spatial domain, it causes minor artifacts in the frequency spectrum as seen in Figure 9. Specifically, frequencies with wavenumber higher than 60 cannot be adequately represented. Quantizing the solution values in spatial domain introduce high-frequency noise which is greater than the original amplitudes. Training the neural surrogates with `float64` precision did not show any performance improvement, besides being significantly more computationally expensive.

**Model architecture**. For all models in Section 4.1, we use the modern U-Net architecture from Gupta et al. [22], which we detail in Table 2. The U-Net consists of an encoder and decoder, which are implemented via several pre-activation ResNet blocks [25, 26] with skip connections between encoder and decoder blocks. The ResNet block is visualized in Figure 10 and consists of Group Normalization [90], GELU activations [27], and convolutions with kernel size 3. The conditioning parameters $\Delta t$ and $\Delta x$ are embedded into feature vector space via sinusoidal embeddings, as for

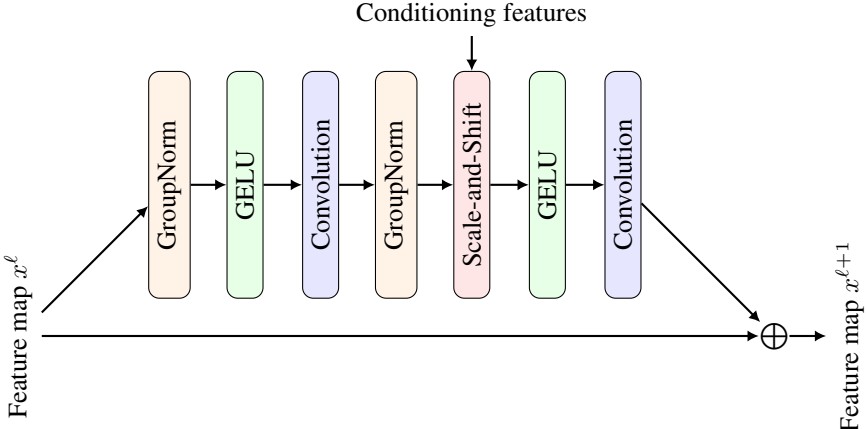

Figure 10: ResNet block of the modern U-Net [22]. Each block consists of two convolutions with GroupNorm and GELU activations. The conditioning features, which are $\Delta t$, $\Delta x$ for the KS dataset and additionally $\nu$ for the parameter-dependent KS dataset, influence the features via a scale-and-shift layer. Residual blocks with different input and output channels use a convolution with kernel size 1 on the residual connection.

Table 3: Hyperparameter overview for the experiments on the KS equation. Hyperparameters have been optimized for the baseline MSE-trained model on the validation dataset, which generally worked well across all models.

| Hyperparameter | Value |
|---|---|
| Input Resolution | 256 |
| Number of Epochs | 400 |
| Batch size | 128 |
| Optimizer | AdamW [52] |
| Learning rate | CosineScheduler(1e-4 $\rightarrow$ 1e-6) |
| Weight Decay | 1e-5 |
| Time step | 0.8s / $4\Delta t$ |
| Output factor | 0.3 |
| Network | Modern U-Net [22] |
| Hidden size | $c_1 = 64$, $c_2 = 128$, $c_3 = 256$, $c_4 = 1024$ |
| Padding | circular |
| EMA Decay | 0.995 |

example used in Transformers [84]. We combine the feature vectors via linear layers and integrate them in the U-Net via AdaGN [61, 64] layers, which predicts a scale and shift parameter for each channel applied after the second Group Normalization in each residual block. We represent it as a 'scale-and-shift' layer in Figure 10. We also experimented with adding attention layers in the residual blocks, which, however, did not improve performance noticeably. The implementation of the U-Net architecture can be found in the public code of Gupta et al. [22].[5]

**Hyperparameters**. We detail the used hyperparameters for all models in Table 3. We train the models for 400 epochs on a batch size of 128 with an AdamW optimizer [52]. One epoch corresponds to iterating through all training sequences and picking 100 random initial conditions each. The learning rate is initialized with 1e-4, and follows a cosine annealing strategy to end with a final learning rate of 1e-6. We did not find learning rate warmup to be needed for our models. For regularization, we use a weight decay of 1e-5. As mentioned in Section 4.1, we train the neural operators to predict 4 time steps ahead via predicting the residual $\Delta u = u(t) - u(t - 4\Delta t)$. For better output coverage of the neural network, we normalize the residual to a standard deviation of about 1 by dividing it with 0.3.

---

[5]`https://github.com/microsoft/pdearena/blob/main/pdearena/modules/conditioned/twod_unet.py`

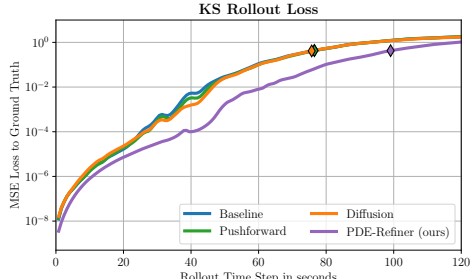 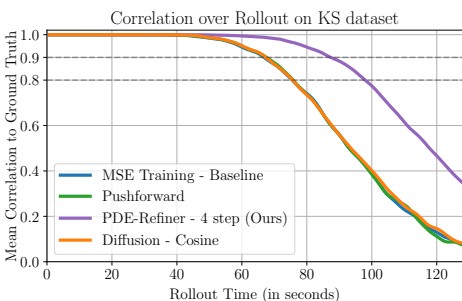

Figure 11: **Left**: Visualizing the average MSE error over rollouts on the test set for four methods: the baseline MSE-trained model (blue), the pushforward trick (green), the diffusion model with standard cosine scheduling (orange), and PDE-Refiner with 8 refinement steps. The markers indicate the time when the method's average rollout correlation falls below 0.8. The y-axis shows the logarithmic scale of the MSE error. While all models have a similar loss for the first 20 seconds, PDE-Refiner has a much smaller increase of loss afterwards. **Right**: showing the average correlation over rollout time. Different thresholds would lead to the same findings in this paper.

Thus, the neural operators predict the next time step via $\hat{u}(t) = u(t - 4\Delta t) + 0.3 \cdot \text{NO}(u(t - 4\Delta t))$. We provide an ablation study on the step size in Appendix E.3. For the modern U-Net, we set the hidden sizes to 64, 128, 256, and 1024 on the different levels, following Gupta et al. [22]. This gives the model a parameter count of about 55 million. Crucially, all convolutions use circular padding in the U-Net to account for the periodic domain. Finally, we found that using an exponential moving average (EMA) [41] of the model parameters during validation and testing, as commonly used in diffusion models [30, 38] and generative adversarial networks [16, 91], improves performance and stabilizes the validation performance progress over training iterations across all models. We set the decay rate of the moving average to 0.995, although it did not appear to be a sensitive hyperparameter.

Next, we discuss extra hyperparameters for each method in Figure 3 individually. The history 2 model includes earlier time steps by concatenating $u(t - 8\Delta t)$ with $u(t - 4\Delta t)$ over the channel dimension. We implement the model with $4\times$ parameters by multiplying the hidden size by 2, i.e. use 128, 256, 512, and 2048. This increases the weight matrices by a factor of 4. For the pushforward trick, we follow the public implementation of Brandstetter et al. [9][6] and increase the probability of replacing the ground truth with a prediction over the first 10 epochs. Additionally, we found it beneficial to use the EMA model weights for creating the predictions, and rolled out the model up to 3 steps. We implemented the Markov Neural Operator following the public code[7] of Li et al. [50]. We performed a hyperparameter search over $\lambda \in \{0.2, 0.5, 0.8\}, \alpha \in \{0.001, 0.01, 0.1\}, k \in \{0, 1\}$, for which we found $\lambda = 0.5, \alpha = 0.01, k = 0$ to work best. The error correction during rollout is implemented by performing an FFT on each prediction, setting the amplitude and phase for wavenumber 0 and above 60 to zero, and mapping back to spatial domain via an inverse FFT. For the error prediction, in which one neural operator tries to predict the error of the second operator, we scale the error back to an average standard deviation of 1 to allow for a better output scale of the second U-Net. The DDPM Diffusion model is implemented using the diffusers library [65]. We use a DDPM scheduler with `squaredcos_cap_v2` scheduling, a beta range of 1e-4 to 1e-1, and 1000 train time steps. During inference, we set the number of sampling steps to 16 (equally spaced between 0 and 1000) which we found to obtain best results while being more efficient than 1000 steps. For our schedule, we set the betas the same way as shown in the pseudocode of Appendix C. Lastly, we implement PDE-Refiner using the diffusers library [65] as shown in Appendix C. We choose the minimum noise variance $\sigma_{\min}^2 = 2\text{e-}7$ based on a hyperparameter search on the validation, and provide an ablation study on it in Appendix E.7.

**Results**. We provide an overview of the results in Figure 3 as table in Table 4. Besides the high-correction time with thresholds 0.8 and 0.9, we also report the one-step MSE error between the prediction $\hat{u}(t)$ and the ground truth solution $u(t)$. A general observation is that the one-step MSE is not a strong indication of the rollout performance. For example, the MSE loss of the history 2 model is twice as low as the baseline's loss, but performs significantly worse in rollout. Similarly,

---

[6]`https://github.com/brandstetter-johannes/MP-Neural-PDE-Solvers/`
[7]`https://github.com/neuraloperator/markov_neural_operator/`

Table 4: Results of Figure 3 in table form. All standard deviations are reported over 5 seeds excluding *Ensemble*, which used all 5 baseline model seeds and has thus no standard deviation. Further, we include the average one-step MSE error of each method on the test set. Notably, lower one-step MSE does not necessarily imply longer stable rollouts (e.g. History 2 versus baseline).

| Method | Corr. $> 0.8$ **time** | Corr. $> 0.9$ **time** | **One-step MSE** |
|---|---|---|---|
| *MSE Training* | | | |
| Baseline | 75.4 $\pm$ 1.1 | 66.5 $\pm$ 0.8 | 2.70e-08 $\pm$ 8.52e-09 |
| History 2 | 61.7 $\pm$ 1.1 | 54.3 $\pm$ 1.8 | 1.50e-08 $\pm$ 1.67e-09 |
| 4$\times$ parameters | 79.7 $\pm$ 0.7 | 71.7 $\pm$ 0.7 | 1.02e-08 $\pm$ 4.91e-10 |
| Ensemble | 79.7 $\pm$ 0.0 | 72.5 $\pm$ 0.0 | 5.56e-09 $\pm$ 0.00e+00 |
| *Alternative Losses* | | | |
| Pushforward [9] | 75.4 $\pm$ 1.1 | 67.3 $\pm$ 1.7 | 2.76e-08 $\pm$ 5.68e-09 |
| Sobolev norm $k = 0$ [50] | 71.4 $\pm$ 2.9 | 62.2 $\pm$ 3.9 | 1.33e-07 $\pm$ 8.70e-08 |
| Sobolev norm $k = 1$ [50] | 66.9 $\pm$ 1.8 | 59.3 $\pm$ 1.5 | 1.04e-07 $\pm$ 3.28e-08 |
| Sobolev norm $k = 2$ [50] | 8.7 $\pm$ 0.9 | 7.3 $\pm$ 0.5 | 7.84e-04 $\pm$ 9.30e-05 |
| Markov Neural Operator [50] | 66.6 $\pm$ 1.0 | 58.5 $\pm$ 2.1 | 2.66e-07 $\pm$ 1.08e-07 |
| Error correction [58] | 74.8 $\pm$ 1.1 | 66.2 $\pm$ 0.9 | 1.46e-08 $\pm$ 1.99e-09 |
| Error Prediction | 75.7 $\pm$ 0.5 | 67.3 $\pm$ 0.6 | 2.96e-08 $\pm$ 2.36e-10 |
| *Diffusion Ablations* | | | |
| Diffusion - Standard Scheduler [30] | 75.2 $\pm$ 1.0 | 66.9 $\pm$ 0.7 | 3.06e-08 $\pm$ 5.24e-10 |
| Diffusion - Our Scheduler | 88.9 $\pm$ 1.0 | 79.7 $\pm$ 1.1 | 2.85e-09 $\pm$ 1.65e-10 |
| *PDE-Refiner* | | | |
| PDE-Refiner - 1 step (ours) | 89.8 $\pm$ 0.4 | 80.6 $\pm$ 0.2 | 3.14e-09 $\pm$ 2.85e-10 |
| PDE-Refiner - 2 steps (ours) | 94.2 $\pm$ 0.8 | 84.2 $\pm$ 0.4 | 5.24e-09 $\pm$ 1.54e-10 |
| PDE-Refiner - 3 steps (ours) | 97.5 $\pm$ 0.5 | 87.0 $\pm$ 0.9 | 5.80e-09 $\pm$ 1.65e-09 |
| PDE-Refiner - 4 steps (ours) | 98.3 $\pm$ 0.8 | 87.8 $\pm$ 1.6 | 5.95e-09 $\pm$ 1.95e-09 |
| PDE-Refiner - 8 steps (ours) | 98.3 $\pm$ 0.1 | 89.0 $\pm$ 0.4 | 6.16e-09 $\pm$ 1.48e-09 |
| PDE-Refiner - 3 steps mean (ours) | 98.5 $\pm$ 0.8 | 88.6 $\pm$ 1.1 | 1.28e-09 $\pm$ 6.27e-11 |

the Ensemble has a lower one-step error than PDE-Refiner with more than 3 refinement steps, but is almost 20 seconds behind in rollout.

As an additional metric, we visualize in Figure 11 the mean-squared error loss between predictions and ground truth during rollout. In other words, we replace the correlation we usually measure during rollout with the MSE. While PDE-Refiner starts out with similar losses as the baselines for the first 20 seconds, it has a significantly smaller increase in loss afterward. This matches our frequency analysis, where only in later time steps, the non-dominant, high frequencies start to impact the main dynamics. Since PDE-Refiner can model these frequencies in contrast to the baselines, it maintains a smaller error accumulation.

**Speed comparison**. We provide a speed comparison of an MSE-trained baseline with PDE-Refiner on the KS equation. We time the models on generating the test trajectories (batch size 128, rollout length $640\Delta t$) on an NVIDIA A100 GPU with a 24 core AMD EPYC CPU. We compile the models in PyTorch 2.0 [62], and exclude compilation and data loading time from the runtime. The MSE model requires 2.04 seconds ($\pm 0.01$), while PDE-Refiner with 3 refinement steps takes 8.67 seconds ($\pm 0.01$). In contrast, the classical solver used for data generation requires on average 47.21 seconds per trajectory, showing the significant speed-up of the neural surrogates. However, it should be noted that the solver is implemented on CPU and there may exist faster solvers for the 1D Kuramoto-Sivashinsky equation.

## D.2 Parameter-dependent KS dataset

**Data generation**. We follow the same data generation as in Appendix D.1. To integrate the viscosity $\nu$, we multiply the fourth derivative estimate $u_{xxxx}$ by $\nu$. For each training and test trajectory, we uniformly sample $\nu$ between 0.5 and 1.5. We show the effect of different viscosity terms in Figure 12.

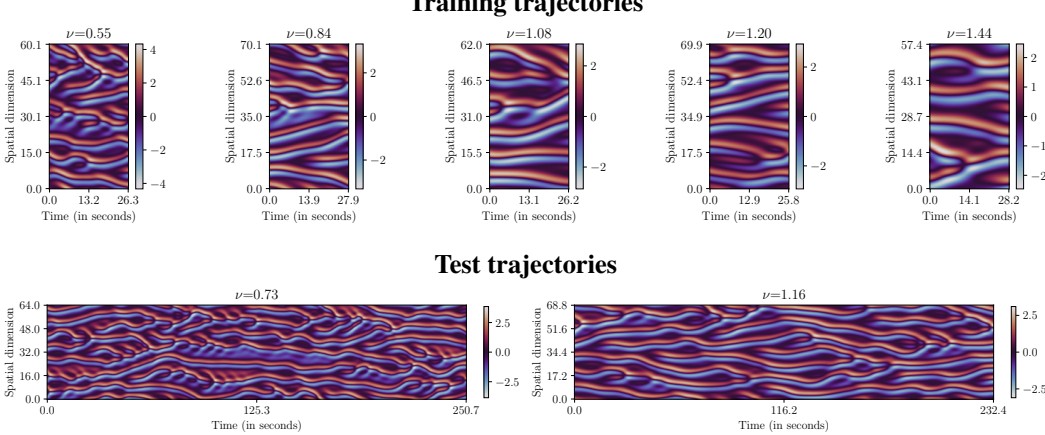

Figure 12: Dataset examples of the parameter-dependent Kuramoto-Sivashinsky dataset. The viscosity is noted above each trajectory. The training trajectories are 140 time steps, while the test trajectories are rolled out for 1140 time steps. Lower viscosities generally create more complex, difficult trajectories.

Table 5: Results of Figure 7 in table form. All standard deviations are reported over 5 seeds.

| Method | Viscosity | Corr. $> 0.8$ time | Corr. $> 0.9$ time |
|---|---|---|---|
| MSE Training | $[0.5, 0.7)$ | $41.8 \pm 0.4$ | $35.6 \pm 0.6$ |
| | $[0.7, 0.9)$ | $57.7 \pm 0.6$ | $50.7 \pm 1.3$ |
| | $[0.9, 1.1)$ | $73.3 \pm 2.3$ | $66.0 \pm 2.5$ |
| | $[1.1, 1.3)$ | $88.0 \pm 1.5$ | $76.7 \pm 2.2$ |
| | $[1.3, 1.5]$ | $97.0 \pm 2.7$ | $85.5 \pm 2.2$ |
| PDE-Refiner | $[0.5, 0.7)$ | $53.1 \pm 0.4$ | $46.7 \pm 0.4$ |
| | $[0.7, 0.9)$ | $71.4 \pm 0.3$ | $64.3 \pm 0.6$ |
| | $[0.9, 1.1)$ | $94.5 \pm 0.6$ | $84.9 \pm 0.6$ |
| | $[1.1, 1.3)$ | $112.2 \pm 0.9$ | $98.5 \pm 1.5$ |
| | $[1.3, 1.5]$ | $130.2 \pm 1.5$ | $116.6 \pm 0.7$ |

**Model architecture**. We use the same modern U-Net as in Appendix D.1. The conditioning features consist of $\Delta t$, $\Delta x$, and $\nu$. For better representation in the sinusoidal embedding, we scale $\nu$ to the range $[0, 100]$ before embedding it.

**Hyperparameters**. We reuse the same hyperparameters of Appendix D.1 except reducing the number of epochs to 250. This is since the training dataset is twice as large as the original KS dataset, and the models converge after fewer epochs.

**Results**. We provide the results of Figure 7 in table form in Table 5. Overall, PDE-Refiner outperforms the MSE-trained baseline by 25-35% across viscosities.

### D.3 Kolmogorov 2D Flow

**Data generation**. We followed the data generation of Sun et al. [79] as detailed in the publicly released code[8]. For hyperparameter tuning, we additionally generate a validation set of the same size as the test data with initial seed 123. Afterward, we remove trajectories where the ground truth solver had NaN outputs, and split the trajectories into sub-sequences of 50 frames for efficient training. An epoch consists of iterating over all sub-sequences and sampling 5 random initial conditions from each. All data are stored in `float32` precision.

---

[8]`https://github.com/Edward-Sun/TSM-PDE/blob/main/data_generation.md`

Table 6: Hyperparameter overview for the experiments on the Kolmogorov 2D flow.

| Hyperparameter | Value |
|---|---|
| Input Resolution | 64×64 |
| Number of Epochs | 100 |
| Batch size | 32 |
| Optimizer | AdamW [52] |
| Learning rate | CosineScheduler(1e-4 $\rightarrow$ 1e-6) |
| Weight Decay | 1e-5 |
| Time step | 0.112s / 16$\Delta t$ |
| Output factor | 0.16 |
| Network | Modern U-Net [22] |
| Hidden size | [128, 128, 256, 1024] |
| Padding | circular |
| EMA Decay | 0.995 |

**Model architecture**. We again use the modern U-Net [22] for PDE-Refiner and an MSE-trained baseline, where, in comparison to the model for the KS equation, we replace 1D convolutions with 2D convolutions. Due to the low input resolution, we experienced that the model lacked complexity on the highest feature resolution. Thus, we increased the initial hidden size to 128, and use 4 ResNet blocks instead of 2 on this level. All other levels remain the same as for the KS equation. This model has 157 million parameters.

The Fourier Neural Operator [49] consists of 8 layers, where each layer consists of a spectral convolution with a skip connection of a $1 \times 1$ convolution and GELU activation [27]. We performed a hyperparameter search over the number of modes and hidden size, for which we found 32 modes with hidden size 64 to perform best. This models has 134 million parameters, roughly matching the parameter count of a U-Net. Models with larger parameter count, e.g. hidden size 128 with 32 modes, did not show any improvements.

**Hyperparameters**. We summarize the chosen hyperparameters in Table 6, which were selected based on the performance on the validation dataset. We train the models for 100 epochs with a batch size of 32. Due to the increased memory usage, we parallelize the model over 4 GPUs with batch size 8 each. We predict every 16th time step, which showed similar performance to models with a time step of 1, 2, 4, and 8 while being faster to roll out. All models use as objective the residual $\Delta u = u(t) - u(t - 16\Delta t)$, which we normalize by dividing with its training standard deviation of 0.16. Thus, we predict the next solution via $\hat{u}(t) = u(t - 16\Delta t) + 0.16 \cdot \text{NO}(...)$. Each model is trained for 3 seeds, and the standard deviation is reported in Table 1.

**Results**. We include example trajectories and corresponding predictions by PDE-Refiner in Figure 13. PDE-Refiner is able to maintain accurate predictions for more than 11 seconds for many trajectories.

**Speed comparison**. All models are run on the same hardware, namely an NVIDIA A100 GPU with 80GB memory and an 24 core AMD EPYC CPU. For the hybrid solvers, we use the public implementation in JAX [6] by Kochkov et al. [43], Sun et al. [79]. For the U-Nets, we use PyTorch 2.0 [14]. All models are compiled in their respective frameworks, and we exclude the compilation and time to load the data from the runtime. We measure the speed of each model 5 times, and report the mean and standard deviation in Section 4.3.

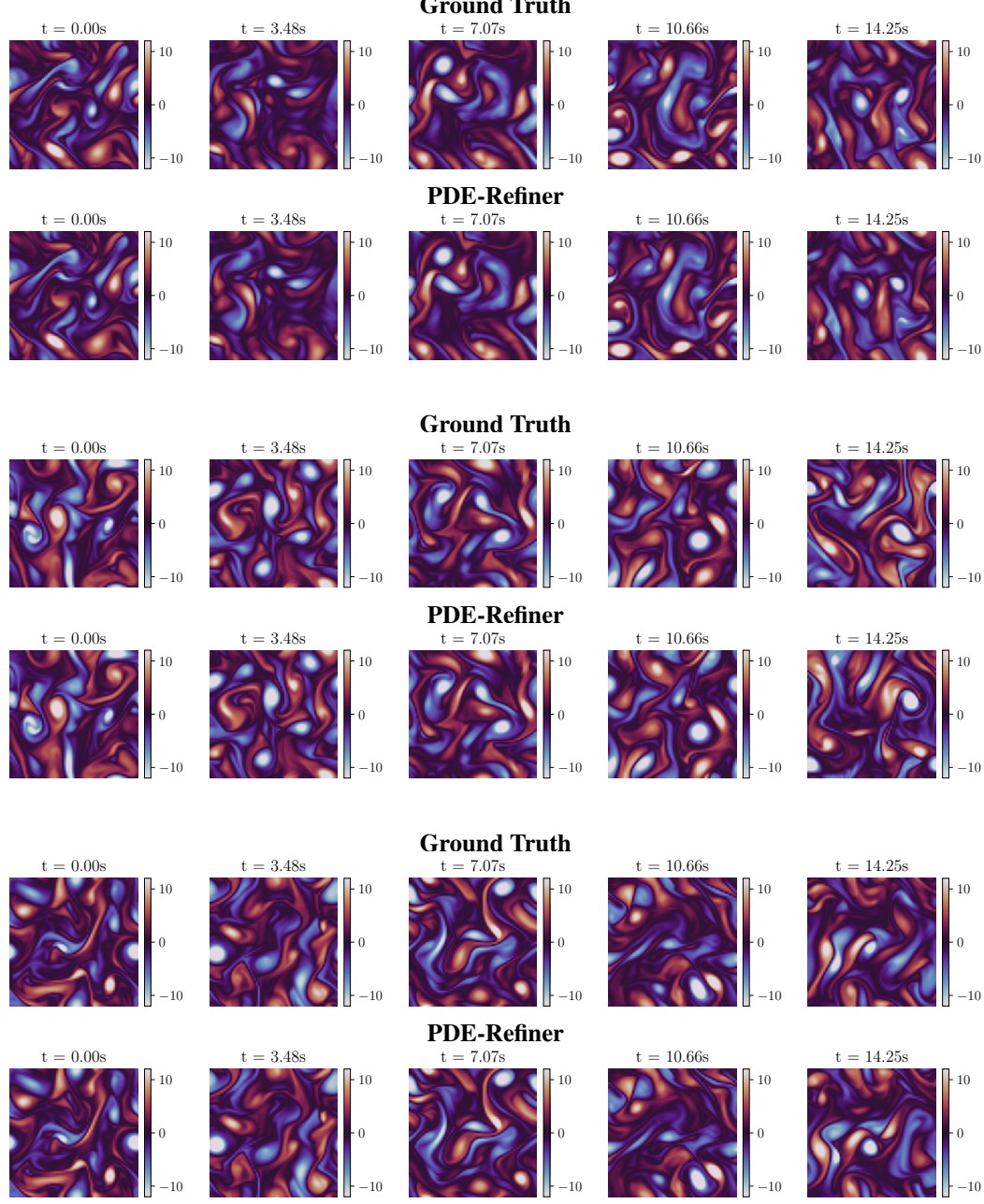

Figure 13: Visualizing the vorticity of three example test trajectories of the 2D Kolmogorov flow, with corresponding predictions of PDE-Refiner. PDE-Refiner remains stable for more than 10 seconds, making on minor errors at 10.66 seconds. Moreover, many structures at 14 seconds are still similar to the ground truth.

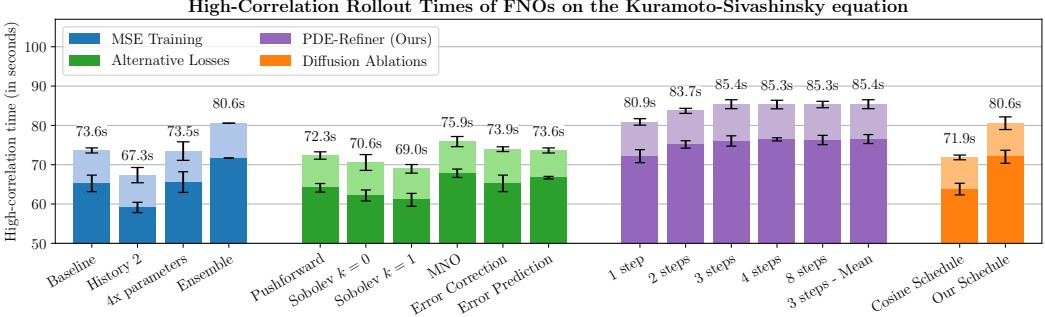

Figure 14: Experimental results of Fourier Neural Operators on the Kuramoto-Sivashinsky equation. All methods from Figure 3 are included here. FNOs achieve similar results as the U-Nets for the baselines. For PDE-Refiner and Diffusion, FNOs still outperforms the baselines, but with a smaller gain than the U-Nets due to the noise objective.

# E  Supplementary Experimental Results

In this section, we provide additional experimental results on the Kuramoto-Sivashinsky equation and the 2D Kolmogorov flow. Specifically, we experiment with Fourier Neural Operators and Dilated ResNets as an alternative to our deployed U-Nets. We provide ablation studies on the predicted step size, the history information, and the minimum noise variance in PDE-Refiner on the KS equation. For the Kolmogorov flow, we provide the same frequency analysis as done for the KS equation in the main paper. Finally, we investigate the stability of the neural surrogates for very long rollouts of 800 seconds.

## E.1  Fourier Neural Operator

Fourier Neural Operators (FNOs) [49] are a popular alternative to U-Nets for neural operator architectures. To show that the general trend of our results in Section 4.1 are architecture-invariant, we repeat all experiments of Figure 3 with FNOs. The FNO consists of 8 layers, where each layer consists of a spectral convolution with a skip connection of a $1 \times 1$ convolution and a GELU activation [27]. Each spectral convolution uses the first 32 modes, and we provide closer discussion on the impact of modes in Figure 15. We use a hidden size of 256, which leads to the model having about 40 million parameters, roughly matching the parameter count of the used U-Nets.

**MSE Training**.  We show the results for all methods in Figure 14. The MSE-trained FNO baseline achieves with 73.6 a similar rollout time as the U-Net (75.4s). Again, using more history information decreases rollout performance. Giving the model more complexity by increasing the parameter count to 160 million did not show any improvement. Still, the ensemble of 5 MSE-trained models obtains a 7-second gain over the individual models, slightly outperforming the U-Nets for this case.

**Alternative losses**.  The pushforward trick, the error correction and the error predictions again cannot improve over the baseline. While using the Sobolev norm losses decrease performance also for FNOs, using the regularizers of the Markov Neural Operator is able to provide small gains. This is in line with the experiments of Li et al. [50], in which the MNO was originally proposed for Fourier Neural Operators. Still, the gain is limited to 3%.

**PDE-Refiner**.  With FNOs, PDE-Refiner again outperforms all baselines when using more than 1 refinement step. The gains again flatten for more than 3 steps. However, in comparisons to the U-Nets with up to 98.5s accurate rollout time, the performance increase is significantly smaller. In general, we find that FNOs obtain higher training losses for smaller noise values than U-Nets, indicating the modeling of high-frequent noise in PDE-Refiner's refinement objective to be the main issue. U-Nets are more flexible in that regard, since they use spatial convolutions. Still, the results show that PDE-Refiner is applicable to a multitude of neural operator architectures.

**Diffusion ablations**.  Confirming the issue of the noise objective for FNOs, the diffusion models with standard cosine scheduling obtain slightly worse results than the baseline. Using our exponential noise scheduler again improves performance to the level of the one-step PDE-Refiner.

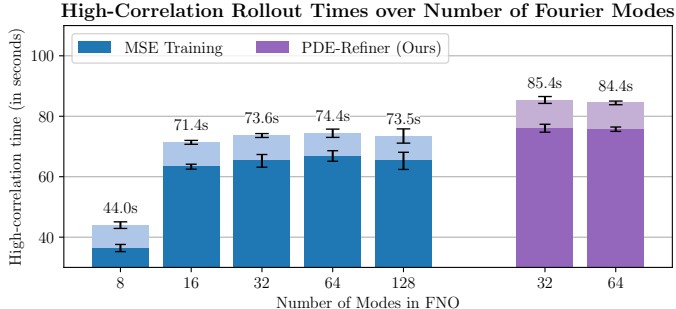

Figure 15: Investigating the impact of the choosing the number of modes in FNOs. Similar to our analysis on the resolution in the U-Nets (Figure 3), we only see minor improvements of using higher frequencies above 16 in the MSE training. Removing dominant frequencies above 8 significantly decreases performance. Similarly, increasing the modes of FNOs in PDE-Refiner has minor impact.

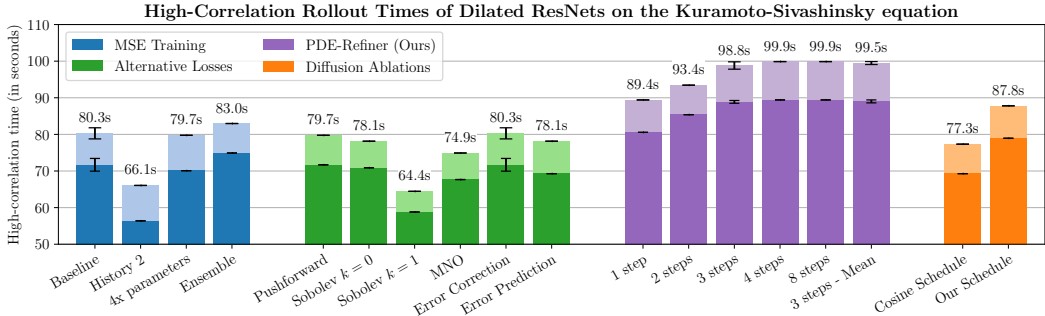

Figure 16: Experimental results of Dilated ResNets on the Kuramoto-Sivashinsky equation. All methods from Figure 3 are included here, with single seeds provided for all methods except the MSE baseline and PDE-Refiner, for which three seeds are shown. The results are overall similar to U-Nets, slightly outperforming the U-Net overall. Yet, PDE-Refiner again outperforms all baselines with a significant margin.

**Number of Fourier Modes**. A hyperparameter in Fourier Neural Operators is the number of Fourier modes that are considered in the spectral convolutions. Any higher frequency is ignored and must be modeled via the residual $1 \times 1$ convolutions. To investigate the impact of the number of Fourier modes, we repeat the baseline experiments of MSE-trained FNOs with 8, 16, 32, 64, and 128 modes in Figure 15. To ensure a fair comparison, we adjust the hidden size to maintain equal number of parameters across models. In general, we find that the high-correlation time is relatively stable for 32 to 128 modes. Using 16 modes slightly decreases performance, while limiting the layers to 8 modes results in significantly worse rollouts. This is also in line with our input resolution analysis of Figure 5, where the MSE-trained baseline does not improve for high resolutions. Similarly, we also apply a 64 mode FNOs for PDE-Refiner. Again, the performance does not increase for higher number of modes.

## E.2 Dilated ResNets

As another strong neural operator, Stachenfeld et al. [78] found Dilated ResNets to perform well on a variety of fluid dynamics. Instead of reducing resolution as in U-Nets, Dilated ResNets make use of dilated convolutions [92] to increase the receptive field of the network and take into account the whole spatial dimension. A Dilated ResNets consists of a ResNet [25] with 4 blocks, each containing 7 convolutional layers with dilation factors [1, 2, 4, 8, 4, 2, 1]. Since the convolutions do not use any stride, all operations are performed on the original spatial resolution, increasing computational cost over e.g. a U-Net with similar parameter count.

For our setup on the KS dataset, we use a Dilated ResNet with channel size 256, group normalization between convolutions and a shift-and-scale conditioning as in the U-Nets. Additionally, we change the default ResNet architecture to using pre-activations [26], allowing for a better gradient flow and

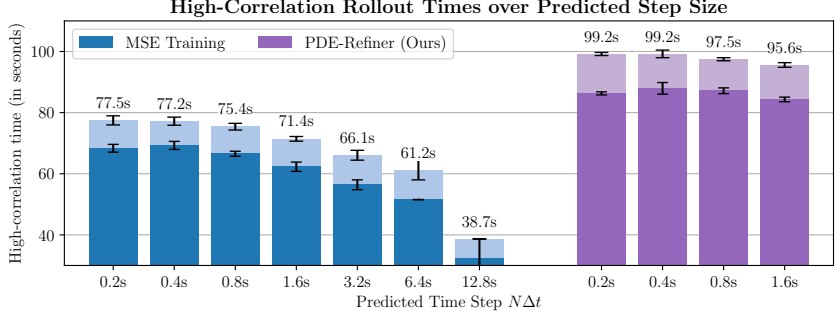

Figure 17: Comparing the accurate rollout times over the step size at which the neural operator predicts. This is a multiple of the time step $\Delta t$ used for data generation (for KS on average 0.2s). For both the MSE Training and PDE-Refiner, lower step size provides longer stable rollouts, where very large time steps show a significant loss in accuracy. This motivates the need for autoregressive neural PDE solvers over direct, long-horizon predictions.

operations on the input directly. While for the MSE baseline, both ResNet version perform equally, we see a considerable improvement for PDE-Refiner, in particular for the refinement steps where early activations can augment the input noise. Overall, the model has around 22 million parameters. More parameters by increasing the channel size did not show to benefit the model.

We show all results in Figure 16. Due to the experiments being computationally expensive, we show most results for a single seed only here, but the trends are generally constant and have not shown to be significantly impacted by standard deviations for other neural operators. Dilated ResNet slightly outperform U-Nets for the MSE baseline, but generally have the same trends across baselines. Furthermore, PDE-Refiner is again able to obtain the longest accurate rollouts, similar to the U-Nets. This verifies the strength of PDE-Refiner and its applicability across various state-of-the-art neural operator architectures.

### E.3 Step Size Comparison

A key advantage of Neural PDE solvers is their flexibility to be applied to various step sizes of the PDEs. The larger the step size is, the faster the solver will be. At the same time, larger step sizes may be harder to predict. To compare the effect of error propagation in an autoregressive solver with training a model to predict large time steps, we repeat the baseline experiments of the U-Net neural operator on the KS equation with different step sizes. The default step size that was used in Figure 3 is 4-times the original solver step, being on average 0.8s. For any step size below 2s, we model the residual objective $\Delta u = u(t) - u(t - \Delta t)$, which we found to generally work better in this range. For any step size above, we directly predict the solution $u(t)$.

**High-correlation time**. We plot the results step sizes between 0.2s and 12.8s in Figure 17. We find that the smaller the step size, the longer the model remains accurate. The performance also decreases faster for very large time steps. This is because the models start to overfit on the training data and have difficulties learning the actual dynamics of the PDE. Meanwhile, very small time steps do not suffer from autoregressive error propagation any more than slightly larger time steps, while generalizing well. This highlights again the strength of autoregressive neural PDE solvers. We confirm this trend by training PDE-Refiner with different step sizes while using 3 refinement steps. We again find that smaller time steps achieve higher performance, and we obtain worse rollout times for larger time steps.

**MSE loss over rollout**. To further gain insights of the impact of different step sizes, we plot in Figure 18 the MSE loss to the ground truth when rolling out the MSE-trained models over time. Models with larger time steps require fewer autoregressive steps to predict long-term into the future, preventing any autoregressive error accumulation for the first step. Intuitively, the error increases over time for all models, since the errors accumulate over time and cause the model to diverge. The models with step sizes 0.2s, 0.4s and 0.8s all achieve very similar losses across the whole time horizon. This motivates our choice for 0.8s as default time step, since it provides a 4 times speedup in comparison to the 0.2s model. Meanwhile, already a model trained with step size 1.6s performs considerable worse in its one-step prediction than a model with step size 0.2s rolled out 8 times. The gap increases

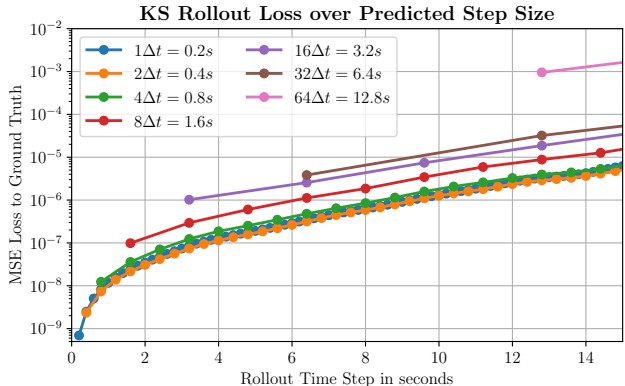

Figure 18: Visualizing the MSE error of MSE-trained models with varying step sizes over the rollout. The models with a step size of $1\Delta t$, $2\Delta t$, and $4\Delta t$ all obtain similar performance. For $8\Delta t$, the one-step MSE loss is already considerably higher than, e.g. rolling out the step size $1\Delta t$ model 8 times. For larger time steps, this gap increases further, again highlighting the strengths of autoregressive solvers.

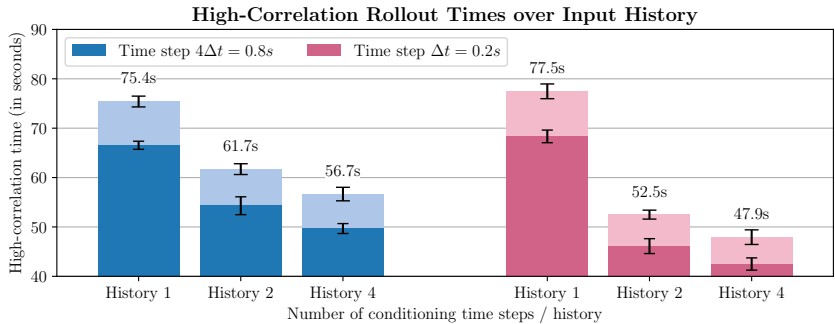

Figure 19: Investigating the impact of using more history / past time steps in the neural operators, i.e., $\hat{u}(t) = \text{NO}(u(t - \Delta t), u(t - 2\Delta t), ...)$, for $\Delta t = 0.8$ and $\Delta t = 0.2$. Longer histories decrease the model's accurate rollout time. This drop in performance is even more significant for smaller time steps.

further the larger the time step becomes. Therefore, directly predicting large time steps in neural PDE solvers is not practical and autoregressive solvers provide significant advantages.

### E.4 History Information

In our experiments on the KS equation, we have observed that using more history information as input decreases the rollout performance. Specifically, we have used a neural operator that took as input the past two time steps, $u(t - \Delta t)$ and $u(t - 2\Delta t)$. To confirm this trend, we repeat the experiments with a longer history of 4 past time steps and for models with a smaller step size of 0.2s in Figure 19. Again, we find that the more history information we use as input, the worse the rollouts become. Furthermore, the impact becomes larger for small time steps, indicating that the autoregressive error propagation becomes a larger issue when using history information. The problem arising is that the difference between the inputs $u(t - \Delta t) - u(t - 2\Delta t)$ is highly correlated with the model's target $\Delta u(t)$, the residual of the next time step. The smaller the time step, the larger the correlation. This leads the neural operator to focus on modeling the second-order difference $\Delta u(t) - \Delta u(t - 2\Delta t)$. As observed in classical solvers [36], using higher-order differences within an explicit autoregressive scheme is known to deteriorate the rollout stability and introduce exponentially increasing errors over time.

We also confirm this exponential increase of error by plotting the MSE error over rollouts in Figure 20. While the history information improves the one-step prediction by a factor of 10, the error of the history 2 and 4 models quickly surpasses the error of the history 1 model. After that, the error of the models continue to increase quickly, leading to an earlier divergence.

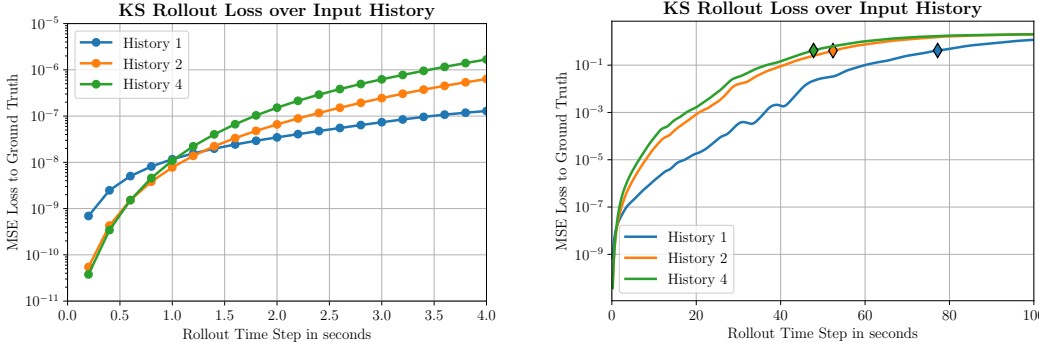

Figure 20: Comparing models conditioned on different number of past time steps on their MSE loss over rollouts. Note the log-scale on the y-axis. The markers indicate the time when the average correlation of the respective model drops below 0.8. The left plot shows a zoomed-in version of the first 4 seconds of the whole 100 second rollout on the right. While using more history information gives an advantage for the first ~5 steps, the error propagates significantly faster through the models. This leads to a significantly higher loss over rollout.

Table 7: Comparing the uncertainty estimate of PDE-Refiner to Input Modulation [5, 71] and Model Ensemble [45, 71] on the MSE-trained models. The metrics show the correlation between the estimated and actual accurate rollout time in terms of the $R^2$ coefficient of determination and the Pearson correlation. PDE-Refiner provides more accurate uncertainty estimates than Input Modulation while being more efficient than an Model Ensemble.

| Method | $R^2$ coefficient | Pearson correlation |
|---|---|---|
| PDE-Refiner | $0.857 \pm 0.027$ | $0.934 \pm 0.014$ |
| Input Modulation [5, 71] | $0.820 \pm 0.081$ | $0.912 \pm 0.021$ |
| Model Ensemble [45, 71] | $0.887 \pm 0.012$ | $0.965 \pm 0.007$ |

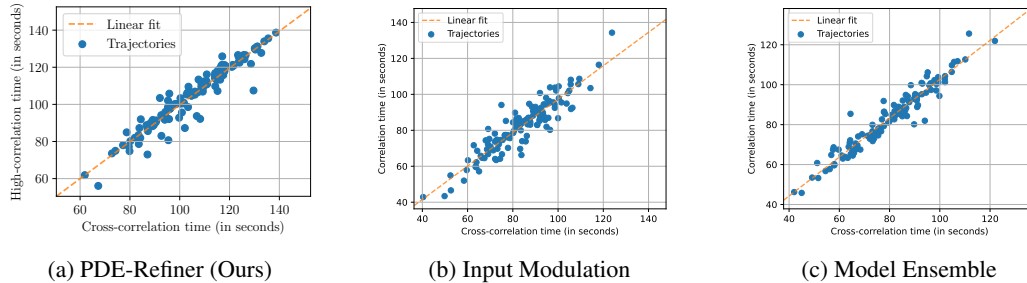

(a) PDE-Refiner (Ours)  (b) Input Modulation  (c) Model Ensemble

Figure 21: Qualitative comparison between the uncertainty estimates of PDE-Refiner, Input Modulation, and the Model Ensemble. Both PDE-Refiner and the Model Ensemble achieve an accurate match between the estimated and ground truth rollout times.

### E.5 Uncertainty Estimation

We extend our discussion on the uncertainty estimation of Section 4.1 by comparing PDE-Refiner to two common baselines for uncertainty estimation of temporal forecasting: Input Modulation [5, 71] and Model Ensemble [45, 71]. Input Modulation adds small random Gaussian noise to the initial condition $u(0)$, and rolls out the model on several samples. Similar to PDE-Refiner, one can determine the uncertainty by measuring the cross-correlation between the rollouts. A Model Ensemble compares the predicted trajectories of several independently trained models. For the case here, we use 4 trained models. For both baselines, we estimate the uncertainty of MSE-trained models as usually applied.

We evaluate the $R^2$ coefficient of determination and the Pearson correlation between the estimated stable rollout times and the ground truth rollout times in Table 7. We additionally show qualitative

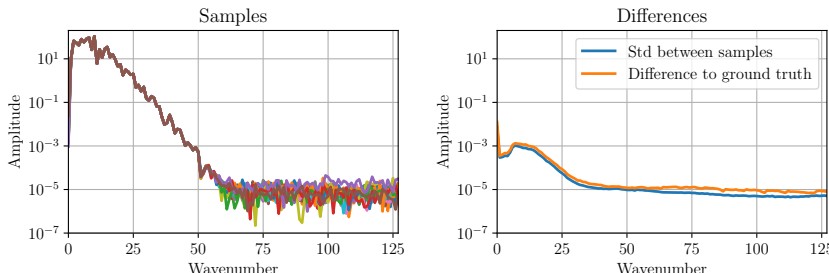

Figure 22: Investigating the spread of samples of PDE-Refiner. The left plot shows the frequency spectrum of 16 samples (each line represents a different sample), with the right plot showing the average difference to the ground truth and to the mean of the samples. The deviation of the samples closely matches the average error, showing that PDE-Refiner adapts its samples to the learned error over frequencies.

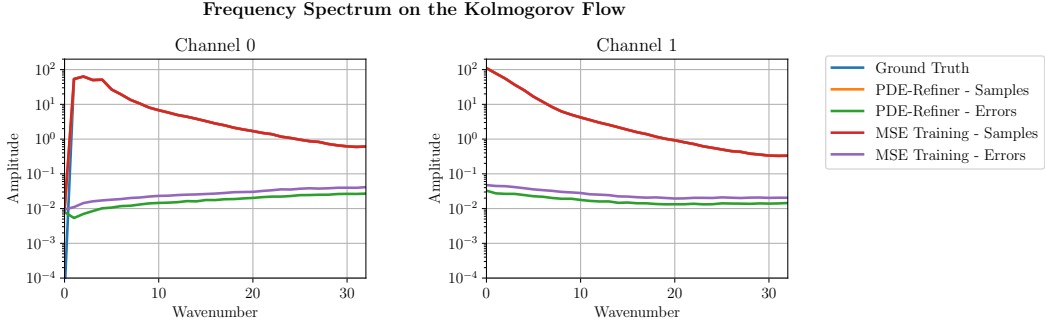

Figure 23: Frequency spectrum on the Kolmogorov Flow. The two plots show the two channels of the Kolmogorov flow. Since the data has a much more uniform support over frequencies than the KS equation, both the MSE-trained model and PDE-Refiner model the ground truth very accurately. Thus, the Ground Truth (blue), PDE-Refiner's prediction (orange) and the MSE-trained prediction (red) overlap in both plots. Plotting the error reveals that PDE-Refiner provides small gains across all frequencies.

results in Figure 21. PDE-Refiner's uncertainty estimate outperforms the Input Modulation approach, showing that Gaussian noise does not fully capture the uncertainty distribution. While performing slightly worse than using a full Model Ensemble, PDE-Refiner has the major advantage that it only needs to be trained, which is particularly relevant in large-scale experiments like weather modeling where training a model can be very costly.

To investigate the improvement of PDE-Refiner over Input Modulation, we plot the standard deviation over samples in PDE-Refiner in Figure 22. The samples of PDE-Refiner closely differs in the same distribution as the actual loss to the ground truth, showing that PDE-Refiner accurately models its predictive uncertainty.

### E.6 Frequency Analysis for 2D Kolmogorov Flow

We repeat the frequency analysis that we have performed on the KS equation in the main paper, e.g. Figure 4, on the Kolmogorov dataset here. Note that we apply a 2D Discrete Fourier Transform and show the average frequency spectrum. We perform this over the two channels of $u(t)$ independently. Figure 23 shows the frequency spectrum for the ground truth data, as well as the predictions of PDE-Refiner and the MSE-trained U-Net. In contrast to the KS equation, the spectrum is much flatter, having an amplitude of still almost 1 at wavenumber 32. In comparison, the KS equation has a more than 10 times as small amplitude for this wavenumber. Further, since the resolution is only $64 \times 64$, higher modes cannot be modeled, which, as seen on the KS equation, would increase the benefit of PDE-Refiner. This leads to both PDE-Refiner and the MSE-trained baseline to model all frequencies accurately. The slightly higher loss for higher frequencies on channel 0 is likely due to missing high-

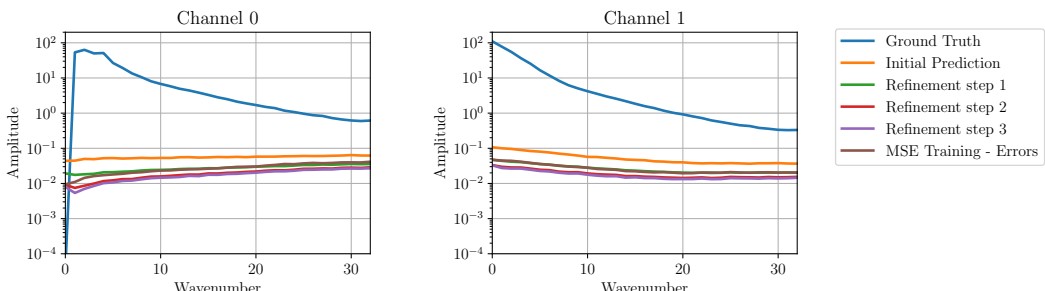

Figure 24: Frequency spectrum of intermediate samples in the refinement process of PDE-Refiner, similar to Figure 4 for the KS equation. The refinement process improves the prediction of the model step-by-step. For the last refinement step, we actually see minor improvements for the lowest frequencies of channel 0. However, due to flatter frequency spectrum, the high frequencies do not improve as much as on the KS equation.

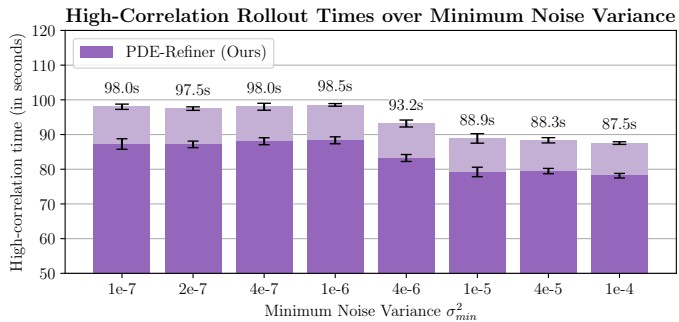

Figure 25: Plotting performance of PDE-Refiner over different values of the minimum noise variance $\sigma_{\min}^2$. Each PDE-Refiner is robust to small changes of $\sigma_{\min}^2$, showing an equal performance in the range of $\left[10^{-7}, 10^{-6}\right]$. Higher standard deviations start to decrease the performance, confirming our analysis of later refinement steps focusing on low-amplitude information. For the experiments in Section 4.1, we have selected $\sigma_{\min}^2 =$ 2e-7 based on the validation dataset.

frequency information, i.e., larger resolution, that would be needed to estimate the frequencies more accurately. Still, we find that PDE-Refiner improves upon the MSE-trained model on all frequencies.

In Figure 24, we additionally plot the predictions of PDE-Refiner at different refinement steps. Similar to the KS equation, PDE-Refiner improves its prediction step by step. However, it is apparent that no clear bias towards the high frequencies occur in the last time step, since the error is rather uniform across all frequencies. Finally, the last refinement step only provides minor gains, indicating that PDE-Refiner with 2 refinement steps would have likely been sufficient.

### E.7 Minimum noise variance in PDE-Refiner

Besides the number of refinement step, PDE-Refiner has as a second hyperparameter the minimum noise variance $\sigma_{\min}^2$, i.e., the variance of the added noise in the last refinement step. The noise variance determines the different amplitude levels at which PDE-Refiner improves the prediction. To show how sensitive PDE-Refiner is to different values of $\sigma_{\min}^2$, we repeat the experiments of PDE-Refiner on the KS equation while varying $\sigma_{\min}^2$. The results in Figure 25 show that PDE-Refiner is robust to small changes of $\sigma_{\min}^2$ and there exist a larger band of values where it performs equally well. When increasing the variance further, the performance starts to decrease since the noise is too high to model the lowest amplitude information. Note that the results on Figure 25 show the performance on the test set, while the hyperparameter selection, in which we selected $\sigma_{\min}^2 =$ 2e-7, was done on the validation set.

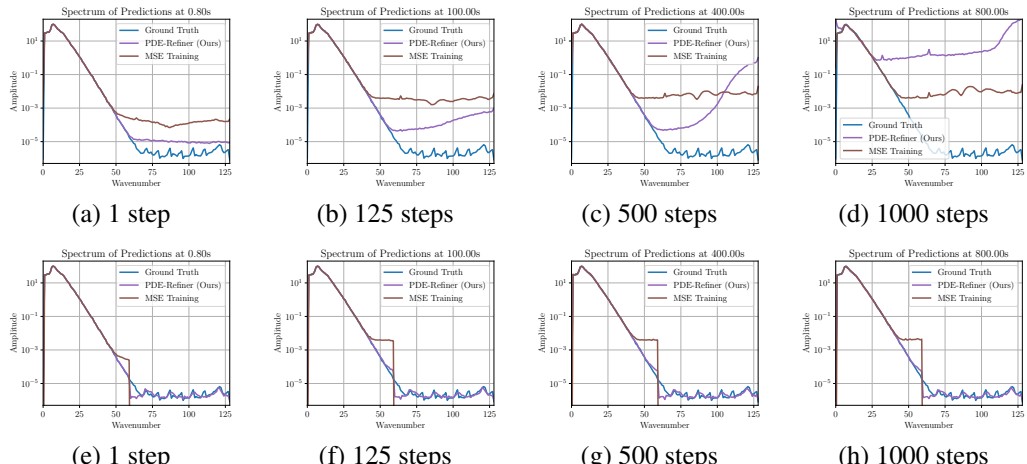

|  |  |  |  |
|:---:|:---:|:---:|:---:|
| (a) 1 step | (b) 125 steps | (c) 500 steps | (d) 1000 steps |
| (e) 1 step | (f) 125 steps | (g) 500 steps | (h) 1000 steps |

Figure 26: Evaluating PDE solver stability over very long rollouts (800 seconds, corresponding to 1000 autoregressive prediction steps). **(a-d)** The frequency spectrum of predictions of an MSE-trained model and PDE-Refiner. Over time, the MSE baseline's overestimation of the high frequencies accumulates. In comparison, PDE-Refiner shows to have an increase of extremely high frequencies, which is likely caused by the continuous adding of Gaussian noise. **(e-h)** When we apply the error correction [58] on our models by setting all frequencies above 60 to zero, PDE-Refiner remains stable even for 1000 steps and does not diverge from the ground truth frequency spectrum.

In combination with the hyperparameter of the number of refinement steps, to which PDE-Refiner showed to also be robust if more than 3 steps is chosen, PDE-Refiner is not very sensitive to the newly introduced hyperparameters and values in a larger range can be considered.

### E.8 Stability of Very Long Rollouts

Besides accurate rollouts, another important aspect of neural PDE solvers is their stability. This refers to the solvers staying in the solution domain and not generating physically unrealistic results. To evaluate whether our solvers remain stable for a long time, we roll out an MSE-trained baseline and PDE-Refiner for 1000 autoregressive prediction steps, which corresponds to 800 seconds simulation time. We then perform a frequency analysis and plot the spectra in Figure 26. We compare the spectra to the ground truth initial condition, to have a reference point of common frequency spectra of solutions on the KS equation.

For the MSE-trained baseline, we find that the high frequencies, that are generally overestimated by the model, accumulate over time. Still, the model maintains a frequency spectrum close to the ground truth for wavenumbers below 40. PDE-Refiner maintains an accurate frequency spectrum for more than 500 steps, but suffers from overestimating the very high frequencies in very long rollouts. This is likely due to the iterative adding of Gaussian noise, that accumulates high-frequency errors. Further, the U-Net has a limited receptive field such that the model cannot estimate the highest frequencies properly. With larger architectures, this may be preventable.

However, a simpler alternative is to correct the predictions for known invariances, as done in McGreivy et al. [58]. We use the same setup as for Figure 3 by setting the highest frequencies to zero. This stabilizes PDE-Refiner, maintaining a very accurate estimation of the frequency spectrum even at 800 seconds. The MSE-trained model yet suffers from an overestimation of the high-frequencies.

In summary, the models we consider here are stable for much longer than they remain accurate to the ground truth. Further, with a simple error correction, PDE-Refiner can keep up stable predictions for more than 1000 autoregressive rollout steps.

