# Supplementary material
# PDE-Refiner: Achieving Accurate Long Rollouts with Neural PDE Solvers

## Table of Contents

## A  Broader Impact

Neural PDE solvers hold significant potential for offering computationally cheaper approaches to modeling a wide range of natural phenomena than classical solvers. As a result, PDE surrogates could potentially contribute to advancements in various research fields, particularly within the natural sciences, such as fluid dynamics and weather modeling. Further, reducing the compute needed for simulations may reduce the carbon footprint of research institutes and industries that rely on such models. Our proposed method, PDE-Refiner, can thereby help in improving the accuracy of these neural solvers, particularly for long-horizon predictions, making their application more viable.

However, it is crucial to note that reliance on simulations necessitates rigorous cross-checks and continuous monitoring. This is particularly true for neural surrogates, which may have been trained on simulations themselves and could introduce additional errors when applied to data outside of its original training distribution. Hence, it is crucial for the underlying assumptions and limitations of these surrogates to be well-understood in applications.

## B  Reproducibility Statement

To ensure reproducibility, we report the used model architectures, hyperparameters, and dataset properties in detail in Section 4 and Appendix D. We additionally include pseudocode for our proposed method, PDE-Refiner, in Appendix C. All experiments on the KS datasets have been repeated for five seeds, and three seeds have been used for the Kolmogorov Flow dataset. Plots and tables with quantitative results show the standard deviation across these seeds.

As existing software assets, we base our implementation on the PDE-Arena [21], which implements a Python-based training framework for neural PDE solvers in PyTorch [60] and PyTorch Lightning [14]. For the diffusion models, we use the library diffusers [63]. We use Matplotlib [33] for plotting and NumPy [85] for data handling. For data generation, we use scipy [81] in the public code of Brandstetter et al. [8] for the KS equation, and JAX [6] in the public code of Kochkov et al. [42], Sun et al. [75] for the 2D Kolmogorov Flow dataset. The usage of these assets is further described in Appendix D. Since our code is proprietary, we include pseudocode in Appendix C, and will release the full code alongside the datasets in this paper upon publication.