# OpenReview forum: "PDE-Refiner: Achieving Accurate Long Rollouts with Neural PDE Solvers"
_NeurIPS.cc/2023/Conference — NeurIPS 2023 spotlight_

### Official Review · Reviewer_xf2d · 2023-06-28

**Soundness:** 4 excellent
**Presentation:** 4 excellent
**Contribution:** 3 good
**Rating:** 8
**Confidence:** 4

**Summary:**

This paper targets the prediction of future PDE states from given initial conditions with deep-learning based surrogate simulators over long rollouts. The authors propose a method titled PDE-Refiner that combines three aspects: 1) autoregressive solution rollouts similar to numerical PDE solvers, 2) direct next-step predictions in a fully data-driven manner (referred to as MSE Training in the paper), and 3) an refinement process to improve the accuracy of predictions based on an iterative denoising objective inspired by diffusion models. Their methodology also links the rollout stability of learned surrogate PDE solvers to the neglect of non-dominant, high spatial frequencies.

The proposed method is evaluated on the 1D Kuramoto-Sivashinsky equation and a 2D Kolmogorov Flow with low spatial resolution. Especially on the former, a large range of different rollout techniques and ablations are provided. The paper also includes a detailed supplemental material that discusses reproducibility, implementation, experimental details, as well as a range of additional evaluations and results.

**Strengths:**

The paper targets the problem of the prediction of PDE solutions which are highly relevant across disciplines like medicine, weather and climate predictions as well as engineering. It explores the benefits of the recently emerging diffusion-like architectures, which have been barely studied in this context and have the potential to improve the speed and accuracy of traditional numerical PDE solvers. To the best of my knowledge the combination of a one-step prediction with an iterative diffusion-based refinement process is a novel approach. Furthermore, the proposed methodology also has the potential to impact other domains, for instance video prediction approaches could benefit from a similar method of next-step predictions followed by iterative refinements.

Clarity and presentation are a clear strength of this work. The methodology and experiment descriptions given in the main paper as well as the appendix are well-written and detailed. Even readers that are not directly familiar with the subtleties of the discussed topics should have no problems following the argumentation of the paper. The usage of mathematical notation is concise without being confusing, and does not interrupt the line of argumentation. Related work is cited appropriately throughout the paper and the usage of existing data sets, architectures and implementations is well documented.

The evaluations in this paper shed light on various design aspects of the proposed architecture and are performed thoroughly, especially so for the 1D cases. A large range of ablations and comparisons to related approaches strengthen the line of argumentation in the text. The appendix contains further results to highlight less crucial aspects. Results are evaluated and discussed thoroughly, and include error bars over multiple random seeds in almost every experiment. In addition, the combination of the provided training details, argumentations for hyperparameter values via ablations and search ranges, and pseudocode (with a promise to release their source code with datasets upon publication) leads to outstanding reproducibility of this work.

Overall, I think this is a high-quality paper that is well-polished and has substantial strengths in terms of contributions, novelty, quality and presentation.


**Weaknesses:**

To me, the result comparison for the 2D Kolmogorov Flow in Table 1 seems problematic. It is clearly mentioned that the values for classical PDE solvers and the hybrid methods are taken from previous work. However, it appears that the correlation computation shown for the ML surrogates is performed differently; thus making it not comparable. Intuitively, I would not expect single-step predictions without refinement on a 64x64 resolution to outperform DNS results computed on a 1024x1024 grid. Furthermore, the comparison with LI-CNN seems to indicate that a major part of the benefit of the proposed method comes from the underlying U-Net architecture instead of the refinement. It would be interesting to see further evaluations to investigate and clarify this behavior.

Another weakness is the choice of generalization tests in the paper. While the different data generation setups describe the test sets clearly, they always come from the exact same domain as the training data (apart from holding out some trajectories with a different random seed). The models do have to generalize to longer trajectories during inference, but are not trained with the full training sequence length as optimization only considers single-step predictions. Especially for simulation data from PDEs, it is very valuable to test model interpolation or extrapolation abilities by holding out certain parameter ranges during training for testing lateron, e.g. different length, viscosity, or forcing values. It would be a highly interesting and promising direction to explore if the findings in terms of rollout stability of this paper hold for more challenging generalization scenarios.

To conclude, this paper has some weaknesses, however addressing them is not immediately crucial as the paper in its current state already provides a large range of solid contributions detailed above.


**Questions:**

1. How are the correlation thresholds of 0.8 / 0.9 (L200) chosen? Is it expected that choosing lower or higher values would substantially change the findings in terms of rollout stability?
2. In the main methodology explanation (around L135) it could be more clear how the cases for k=0 and k!=0 are balanced. Furthermore, it is not directly obvious that the same model is used to learn k=0 and k!=0. However, this is very clear with the explanations and pseudocode in the appendix.


**Limitations:**

While there is a short discussion on interpreting different diffusion samples in terms of uncertainty estimation (L272), the aspect of posterior sampling is not discussed. Traditional diffusion models allow for generating substantially different samples from the target distribution, but the proposed method is significantly limited in that regard. As the base of the PDE-Refiner is a standard one-step prediction, the diffusion-based refinement process is not able to generate meaningful differences in the samples anymore. In my opinion, this aspect should be discussed in the paper.

I think it would be important to clearly mention the number of rollout steps for each setup directly. Delta t from the simulation is given first, and that the model takes a lot fewer rollout steps is not immediately obvious (L187 and L307). Especially so for the 2D case where the overall number of network rollout steps is not given and can only roughly be inferred via 16 delta t and the t=14.25 step in Figure 12 in the appendix.

Otherwise, limitations of the proposed approach are discussed with sufficient detail, but it could be mentioned that the findings only hold for data from the training domain as mentioned above.

---

> ### Author Rebuttal · Authors · 2023-08-09
>
> Thank you for your encouraging review and thoughtful feedback. We provide answers to your questions and concerns below.
>
> **2D Kolmogorov Flow - the correlation computation shown for the ML surrogates is performed differently**
>
> The evaluation for the 2D Kolmogorov Flow (lines 328-333) follows the same setup as Sun et al. [2023], from which we take the related results of DNS and hybrid solver. We use the same evaluation code and datasets to ensure fair comparisons. If there is any unclarity in whether we evaluated differently, we would be happy to clarify it.
>
> **I would not expect single-step predictions without refinement on a 64x64 resolution to outperform DNS results computed on a 1024x1024 grid.**
>
> DNS is a general PDE solver, while the neural solver have been specifically trained for this dataset. Certain elements of the PDE such as the equation, the Reynolds number and structure of the initial conditions (Gaussian blobs) are fixed. Hence, neural models can specialize on this data and interpolate the data very well from low resolution, as shown in Sun et al. [2023]. Similarly, the best hybrid approach (TSM) at a resolution of 64x64 can outperform DNS at 1024x1024 as well. We use this dataset since it is a common benchmark, especially for hybrid and neural methods, but a larger, diverse 2D study across hybrid and neural solvers would be interesting for future work.
>
> **The comparison with LI-CNN seems to indicate that a major part of the benefit of the proposed method comes from the underlying U-Net architecture instead of the refinement.**
>
> We agree that scaling up the networks for hybrid methods, especially learned correction, could improve their performance closer to the neural methods. However, this increases computation time, since we need to execute both the classical solver *and* the network. Already for the smaller network used in the hybrid solvers, PDE-Refiner and the classical U-Net operator are faster than the hybrid solvers. Yet, adding the refinement to the U-Net significantly improves the rollout performance, highlighting the benefit of PDE-Refiner also in this setting.
>
> **Test sets always come from the exact same domain as the training data. [...] It would be a highly interesting and promising direction to explore if the findings in terms of rollout stability of this paper hold for more challenging generalization scenarios.**
>
> This would be indeed interesting to investigate. In the current setup of the KS dataset, the models need to generalize to unseen time domains, because all training trajectories are sampled between 65s and 110s after the initial condition, while test trajectories reach until 220s. Given our parameter-dependent dataset, we plan to run a study on training models on viscosities between 0.7 and 1.3 to evaluate their extrapolation to 0.5-0.7 and 1.3-1.5, and training on the ranges 0.5-0.8 and 1.2-1.5 for testing interpolation to 0.8-1.2. Due to the limited rebuttal time, we can't report the results yet and will add them for the final paper version. Further, based on your suggestion, we will mention in the limitations that most of our study focuses on the same train-test domains.
>
> **How are the correlation thresholds of 0.8 / 0.9 (L200) chosen? Is it expected that choosing lower or higher values would substantially change the findings in terms of rollout stability?**
>
> The thresholds were chosen to align with prior work (Sun et al. [2023], Kochkov et al. [2021]) and represent time points at which the solution starts to visually slightly differ from the ground truth. In the rebuttal Figure 2, we plot the correlation over rollout time for different methods. For any reasonable threshold, the findings remain the same. The metric closely aligns with when the prediction exceeds a certain MSE loss, as shown in Fig 10 in Appendix D.1.
>
> **It could be more clear how the cases for $k=0$ and $k\neq 0$ are balanced. Furthermore, it is not directly obvious that the same model is used to learn $k=0$ and $k\neq 0$. However, this is very clear with the explanations and pseudocode in the appendix.**
>
> Thank you for raising this point of unclarity. For efficiency, we indeed use the same model for all steps, including $k=0$ and $k\neq 0$, and uniformly sample them during training. Experiments with weighing the cases differently did not improve results. We will clarify this more in the main text.
>
> **[...] the aspect of posterior sampling is not discussed. Traditional diffusion models allow for generating substantially different samples from the target distribution, but the proposed method is significantly limited in that regard.**
>
> Agreed, the main goal of PDE-Refiner is not to generate highly diverse samples, but enable accurate modeling of low-amplitude frequencies. In Appendix E.4, we analyze the sample distribution of PDE-Refiner, and find that samples differ between each other in the same spectrum structure as to the ground truth, which makes them useful for estimating uncertainty. Still, we will highlight this limitation better in the final version of the paper.
>
> **I think it would be important to clearly mention the number of rollout steps for each setup directly.**
>
> That's a good point. We initially didn't report the rollout steps since the time step itself (i.e. seconds) had a much greater impact. Smaller time steps require many more autoregressive steps, but yet perform better as shown in Appendix E.2. For the presented results on KS dataset, the MSE baseline stays accurate for 94 rollout steps, and PDE-Refiner for 122 steps. For the 2D dataset, 85 for the baseline and 95 for PDE-Refiner. We will add this to the paper.
>
> ---
>
> [Kochkov et al., 2021] Kochkov, Dmitrii, et al. Machine learning–accelerated computational fluid dynamics. Proceedings of the National Academy of Sciences 118.21 (2021).
>
> [Sun et al., 2023] Sun, Zhiqing, et al. A Neural PDE Solver with Temporal Stencil Modeling. ICML (2023).

---

> > ### Comment · Reviewer_xf2d · 2023-08-17
> > **Reply**
> >
> > I thank the authors for the comments and clarifications, I still support an “accept” for this paper.

---

### Official Review · Reviewer_JgV1 · 2023-07-05

**Soundness:** 3 good
**Presentation:** 2 fair
**Contribution:** 3 good
**Rating:** 6
**Confidence:** 4

**Summary:**

This paper focuses on the long-term simulation of time-dependent PDEs. The authors find that high frequencies in solutions are always neglected, impeding the long-term rollout performance. This paper presents the PDE-Refiner by introducing the diffusion model to refine the model prediction. PDE-Refiner can boost the model performance consistently in representative fluid dynamics benchmarks.

**Strengths:**

1. This paper presents an interesting finding that the key problem in fluid simulation as the prediction of high frequencies.

2. The proposed methods can consistently boost the model prediction.

3. Detailed ablations are provided.

**Weaknesses:**

1. About the model performance.

From Figure 1(b), we can find that the PDE-Refiner will smooth the amplitude of high frequencies. This should be discussed as a limitation.

2. Compare with a simple baseline.

There is a simple baseline to enhance the model performance in high frequencies. That is calculating MSE loss in the frequency domain. Concretely, calculating MSE between the spectrum of ground truth and model prediction.

3. The diffusion model design can make the method data-hungry.

It is common knowledge in deep learning community that diffusion models are data-hungry. Thus, it is necessary to compare the performance of UNet and PDE-Refiner-UNet under the limited data setting (ranging from 10% data to 100% data).

4. Apply PDE-Refiner to more advanced neural operators.

Many recent models perform well in long-term rollout with only MSE loss. Given the authors only apply PDE-Refiner to U-Net, it is necessary to check whether the proposed method can further boost advanced non-FNO models, such as LSM [1].

[1] Solving High-Dimensional PDEs with Latent Spectral Models, ICML 2023

5. Concerns about the model efficiency.

As the authors pointed out, the PDE-Refiner is time-consumption. Besides, the authors should also compare the GPU memory. I think the poor efficiency can seriously affect the model practicability.

**Questions:**

Since the computation efficiency is the major problem of this method, is there some potential solution to speed up the PDE-Refiner efficiency?

**Limitations:**

They have discussed limitations.

---

> ### Author Rebuttal · Authors · 2023-08-09
>
> Thank you for your valuable review and helpful suggestions, especially the question on the low data regime was very interesting. We respond to your questions and concerns below.
>
> **1. From Figure 1(b), we can find that the PDE-Refiner will smooth the amplitude of high frequencies. This should be discussed as a limitation.**
>
> This is a good catch, and actually an imprecision in the figure. We will clarify in the paper that we show the *average* over samples for PDE-Refiner, while showing a single example for the ground truth. We visualize individual samples in Figure 20 (Appendix E.4), showing that applying PDE-Refiner does not oversmooth these frequencies. We also point out that the frequencies above $\sim 60$ are only looking so chaotic in the GT data due to float32 precision limitation. See Figure 8 (Appendix D.1) for how the amplitude continues to decrease for float64.
>
> **2. There is a simple baseline to enhance the model performance in high frequencies. That is calculating MSE loss in the frequency domain. Concretely, calculating MSE between the spectrum of ground truth and model prediction.**
>
> This corresponds to the Sobolev norm $k=0$ baseline in our experiments (e.g. Figure 1, main paper), calculating the MSE between the predicted and GT spectrum. We do not observe any improvements over MSE baseline when using Sobolev norm, since the amplitudes of the high frequencies are still by several magnitudes smaller than the dominant frequencies. Thus, the loss focuses on the same frequencies as the regular MSE loss. Similarly, the Sobolev norm $k=1$ setup gives the high frequencies a higher weight in the loss, but results in worse modeling of the lower frequencies and hence again shorter rollout time.
>
> **3. It is common knowledge in deep learning community that diffusion models are data-hungry. Thus, it is necessary to compare the performance of UNet and PDE-Refiner-UNet under the limited data setting (ranging from 10$\%$ to 100$\%$ data).**
>
> This has become a valuable input to us. We first point out that we are not training the diffusion model on a complex multi-modal distribution like unconditional image generation, but rather on a deterministic target. Hence, we don't expect PDE-Refiner to be more data hungry. Based on your suggestion, we verify this by training PDE-Refiner and an MSE baseline on 10$\%$, 20$\%$ and 50$\%$ of the training trajectories in the KS dataset. The results in the rebuttal Figure 4 show that PDE-Refiner consistently outperforms the baseline in all settings, with PDE-Refiner at 10\% of the data performing on par to the MSE model at 100\%. Moreover, the relative improvement of PDE-Refiner increases to 50$\%$ for the low data regime, since denoising data has an ever-changing objective and input, effectively acting as a *data augmentation*. Thank you for this input, which highlights another aspect of PDE-Refiner, which we are happy to discuss in the main paper.
>
> **4. Many recent models perform well in long-term rollout with only MSE loss. Given the authors only apply PDE-Refiner to U-Net, it is necessary to check whether the proposed method can further boost advanced non-FNO models, such as LSM [1].**
>
> Fully agreed. We have added FNO studies (matching the U-Net ones in the main paper) in Fig 13, and FNO ablations in Fig 14 in the submitted appendix. Additionally, in the rebuttal Figure 1, we have added results for Dilated ResNets, in which PDE-Refiner shows similar gains as on U-Nets. U-Nets, FNOs, and Dilated ResNets are the best performing models in both comprehensive overviews of Gupta \& Brandstetter [2022] and Stachenfeld et al. [2022], highlighting PDE-Refiners applicability across state-of-the-art neural operators. We are happy to discuss more settings for the final paper, such as the newly proposed LSM, which was unfortunately not feasible in the short time frame of the rebuttal.
>
> **5. As the authors pointed out, the PDE-Refiner is time-consumption. Besides, the authors should also compare the GPU memory.**
>
> In terms of memory, we only add a tensor of the input size for the noise and minor computations for the conditioning on refinement step index $k$. Note that during training, we sample the index $k$ and do not require the whole refinement process to be performed. As listed in the rebuttal Table 1, the differences in terms of GPU memory between PDE-Refiner and the common MSE objective are minor (below 1\%) in both training and testing.
>
> **Since the computation efficiency is the major problem of this method, is there some potential solution to speed up the PDE-Refiner efficiency?**
>
> As shown in experiments, PDE-Refiner can gain higher accuracy than models with 4x the parameters, therefore allowing PDE-Refiner to run smaller, more efficient networks than standard neural operators at the same accuracy. Furthermore, early steps require lower accuracy, such that future work could investigate smaller networks and adaptive computation for different steps. Finally, for training efficiency, we share the same neural operator between refinement steps. Training a separate network per step would allow for smaller networks and fewer steps, increasing inference efficiency.
>
> ---
>
> [Gupta \& Brandstetter, 2022] Gupta, Jayesh K., and Johannes Brandstetter. Towards multi-spatiotemporal-scale generalized pde modeling. arXiv preprint arXiv:2209.15616 (2022).
>
> [Stachenfeld et al., 2022] Stachenfeld, Kimberly, et al. Learned coarse models for efficient turbulence simulation. ICLR (2022).

---

> > ### Comment · Reviewer_JgV1 · 2023-08-14
> > **Thanks for your response but the model generality concern still remains.**
> >
> > Thanks for your response. My concerns about the efficiency and limited data performance have been resolved.
> >
> > However, I still expect the model performance in more advanced non-FNO models, since the relative promotion can be significant in classical methods but degenerate in state-of-the-art models.
> >
> > Actually, FNO is a common-used baseline. A lots of models surpass FNO:
> > - MWT: Multiwavelet-based Operator Learning for Differential Equations, NeurIPS 2021
> > - U-NO: U-shaped Neural Operators, arXiv 2022
> > - LSM: Solving High-Dimensional PDEs with Latent Spectral Models, ICML 2023
> >
> > I cannot confirm the model effectiveness without the experiments on these advanced models.

---

> > > ### Author Response · Authors · 2023-08-14
> > > **We are running more experiments**
> > >
> > > We have already started LSM runs over the weekend, and are now starting U-NO runs. So far, LSM runs show the same pattern as reported in the paper + appendix for all other architectures and for all experimental settings. We are posting the results here as soon as they are finished.

---

> > > > ### Author Response · Authors · 2023-08-16
> > > > **New results on LSM and U-NO models -- model generality concern addressed**
> > > >
> > > > As suggested by the reviewer, we have performed additional experiments with UNO [Rahman et al., 2023] and LSM [Wu et al., 2023], using both the common MSE loss and PDE-Refiner. We base our implementation of the methods on the code from the official repositories of the respective papers. We apply the models to the KS dataset by transferring their operations (convs, FFT, attention, etc.) to 1D and implement the conditioning on the equation parameters ($\Delta t, \Delta x$) and $k$ for PDE-Refiner using scale-and-shift layers, as done for the other operators. We scale the models to a similar parameter count as the other models ($\sim$50M), resulting in 45.7M parameters for UNO and 56.1M for LSM. After an initial hyperparameter tuning, we obtained the following results over three seeds:
> > > >
> > > > | **Model**                  | **Corr. $>0.8$**  | **Corr. $>0.9$** |
> > > > |---------------------------|------------------|------------------|
> > > > | UNO - MSE                 | $60.7s \pm 1.9s$ | $51.1s \pm 1.4s$ |
> > > > | UNO - MSE $4\times$params | $62.0s \pm 0.8s$ | $52.9s \pm 1.1s$ |
> > > > | UNO - PDE-Refiner         | $\mathbf{78.2}s \pm 1.2s$ | $\mathbf{70.3}s \pm 0.8s$ |
> > > > |---------------------------|------------------|------------------|
> > > > | LSM - MSE                 | $66.1s \pm 2.1s$ | $57.2s \pm 1.8s$ |
> > > > | LSM - MSE $4\times$params | $66.3s \pm 3.3s$ | $56.9s \pm 2.5s$ |
> > > > | LSM - PDE-Refiner         | $\mathbf{81.8}s \pm 0.8s$ | $\mathbf{72.6}s \pm 0.5s$ |
> > > >
> > > > We find that for both UNO and LSM, PDE-Refiner again provides a significant improvement over the MSE baseline. Following the trend of the other operators, even models with 4 times the parameter count fall similarly behind PDE-Refiner. More extensive hyperparameter tuning could slightly improve the operators' performance further. This would benefit the MSE model and PDE-Refiner equally, and hence keep the trend the same.
> > > >
> > > > In conclusion, the experiments highlight the wide applicability of PDE-Refiner to various architectures, having shown considerable improvements across five commonly used SOTA neural operators. We therefore have addressed the model generality concern and have applied PDE-Refiner to 5 different architectures now.
> > > >
> > > > Thank you again for your effort in reviewing the paper! Please let us know if you have any further questions or concerns.
> > > >
> > > > ---
> > > >
> > > > [Rahman et al., 2023] Rahman, M. A., Ross, Z. E., and Azizzadenesheli, K. (2023). U-no: U-shaped neural operators. TLMR.
> > > >
> > > > [Wu et al., 2023] Wu, H., Hu, T., Luo, H., Wang, J. and Long, M. (2023). Solving High-Dimensional PDEs with Latent Spectral Models. ICML 2023

---

> > > > > ### Comment · Reviewer_JgV1 · 2023-08-16
> > > > >
> > > > > Thanks for the author's response and supplemented experimental results. The concern about model generality has been resolved. Thus, I raise my score to 6.

---

### Official Review · Reviewer_YYxA · 2023-07-06

**Soundness:** 3 good
**Presentation:** 4 excellent
**Contribution:** 3 good
**Rating:** 7
**Confidence:** 3

**Summary:**

This work introduces a PDE-Refiner model to address the challenge of accurately predicting the process of time-dependent partial differential equations (PDEs) over long rollouts. The model can be divided in two parts: prediction (based on FNO or UNet) and multi-step refinement (based on the diffusion model). By refining the prediction that includes non-dominant spatial frequency information, the model outputs stability and accuracy during rollouts. The model is validated on 1D Kuramoto-Sivashinsky (KS) and 2D Kolmogorov equations.

**Strengths:**

1. Good presentation: the categories of recent works are clear, and the challenges of accurate long rollouts description are easy to follow. The methodology is constructed in an easy-read way.
2. Analysis of the long-term prediction error based on the frequency spectrum is interesting and can inspire future research.
3. Using the diffusion model to deal with low-amplitude error in PDE settings is natural and reasonable and novel to the best of my knowledge.

**Weaknesses:**

1. The statement “the primary errors in predicting a one-step solution arise from inaccuracies in modeling the dynamics of these low frequencies.” comes from the observation of simulation results in the 1D KS equation only. However, no further analytical support was provided for other settings, such as in the 2D cases.
2. Even though the evaluation speed is acknowledged by the author as a limitation, the duration of high correlation improvements ($10.659$ of the proposed model v.s. $9.663$ for the MSE training with UNet) shown in Table 1 could be considered as minor, given the inference time is $4 \times$ longer. It is not clear whether the proposed approach is Pareto efficient in terms of the speed / accuracy tradeoff.

**Questions:**

1. As introduced, there are three categories of PDE surrogates. The U-Net and FNO are neural approaches. Can your method be applied to hybrid methods as well?
2. Why is the duration of the high correlation of DNS methods shorter than the neural model, even in the same resolution?

**Limitations:**

Major limitations were covered in the paper.

---

> ### Author Rebuttal · Authors · 2023-08-09
>
> Thank you for helpful review and thoughtful feedback. Please find below our response to your questions and concerns raised.
>
> **Analytical support for 2D case for the statement “the primary errors in predicting a one-step solution arise from inaccuracies in modeling the dynamics of these low frequencies.”**
>
> We have provided a similar analysis to the main paper for the 2D Kolmogorov flow in Appendix E.5. The statement generally holds for moderate/high-resolution inputs, and typically biased frequency spectra. The 2D Kolmogorov dataset is low resolution and a bit less biased, so effects are not as strong as for the 1D Kuramoto-Sivashinsky equation, but still low frequencies are dominant over high frequencies.
>
> **It is not clear whether the proposed approach is Pareto efficient in terms of the speed / accuracy tradeoff - Table 1.**
>
> This is a fair point. We argue that PDE-Refiner allows for a flexible selection of speed vs accuracy tradeoff. For the current state-of-the-art, it provides a closer to Pareto efficient solution, since we showed in Figure 3 that increasing the parameter size is not helping for standard MSE models (which would effectively also require 4x more compute). And thus, even with 4x longer inference, it is faster than current hybrid approaches and more than 100x faster than the classical solver. We finally argue that the presented 2D Kolmogorov dataset is already quite exhaustively studied and the obtained correlation improvements are significant.
>
> **Can PDE-Refiner be applied to hybrid methods as well?**
>
> The refinement process of PDE-Refiner could be applied to learned correction (LC) [Kochkov et al., 2021] as well, since the correction model will suffer from the same biases. Other hybrid methods aim to predict parameters within classical solver, which do not really fit in the framework and may not suffer from the biases as much as plain neural approaches.
>
> **Why is the duration of the high correlation of DNS methods shorter than the neural model, even in the same resolution?**
>
> The ground truth data is DNS generated at a resolution of 2048x2048. Running DNS on smaller resolution will naturally miss data which is difficult to interpolate for these methods, and thus diverge from the ground truth earlier. Neural methods are able to predict the missing high-resolution information much better than simple interpolation, as shown in Sun et al. [2023] on this dataset. This is since certain elements of the PDE such as the equation, the Reynolds number and structure of the initial conditions (Gaussian blobs) are fixed. Thus, neural models can outperform DNS on various resolutions.
>
> ---
>
> [Kochkov et al., 2021] Kochkov, Dmitrii, et al. Machine learning–accelerated computational fluid dynamics. Proceedings of the National Academy of Sciences 118.21 (2021).
>
> [Sun et al., 2023] Sun, Zhiqing, et al. A Neural PDE Solver with Temporal Stencil Modeling. ICML (2023).

---

> > ### Comment · Reviewer_YYxA · 2023-08-16
> > **Reply**
> >
> > Thank you for your answers!
> >
> > I have read your answers as well as the points raised by the other reviewers. My main concerns have been solved.
> > I think your work is a solid contribution to the area of Scientific Machine Learning and can inspire several future works. As such, I would recommend your paper for acceptance and have raised my score.

---

### Official Review · Reviewer_K6r8 · 2023-07-06

**Soundness:** 2 fair
**Presentation:** 3 good
**Contribution:** 3 good
**Rating:** 6
**Confidence:** 2

**Summary:**

Recently, there has been active research on using deep neural networks as surrogates to predict the temporal evolution of solutions in time-dependent partial differential equations (PDEs). Improving the accuracy of long-term predictions for surrogate models that take previous time steps' solutions as inputs and predict the solution at the next time step is a critical challenge. In this paper, the authors propose a PDE-Refiner to address this issue and enable accurate prediction of solution over long time periods.

**Strengths:**

In addressing the issues arising from using MSE loss to predict the solution $u(t)$ at the next time step based on the previous solution $u(t-\Delta t)$, the authors conducted an analysis of the amplitude at different frequencies (Figure 1). To overcome these challenges, they introduced the denoising idea and developed the PDE-Refiner, enabling iterative prediction of the long-term evolution of solutions, $u(t)$. The various experiments in Section 4 confirmed the effectiveness of the PDE-Refiner in capturing high-correlation time over extended periods.

**Weaknesses:**

I appreciate the thorough review of the paper. It seems that there are some confusing aspects that could benefit from further clarification. Specifically, the relationship between denoising and the diverse amplitudes of frequencies in the context of enabling long rollouts for PDE solution prediction is not well-explained. Sections 3 and 3.1 touch on this topic, but it would be helpful to bridge the gap and provide a clearer explanation of how the introduced Gaussian noise, when added and subtracted, enables denoising across various sigma values, thus facilitating long rollouts for different frequency amplitudes. This clarification would greatly assist in understanding the interrelation between these concepts.

Furthermore, I am curious to know how the proposed PDE-Refiner, in combination with other models such as FNO and Graph Neural Networks (GNN) that predict the next time step based on previous time steps, improves accuracy for long-term predictions compared to existing models. Although it is mentioned that combining FNO poses challenges due to the elimination of high frequencies, I believe that using the learned parameters from all frequency domains in FNO could mitigate this issue. It would be valuable to see experimental results showcasing the improved accuracy for longer time predictions when applying PDE-Refiner to models like FNO and graph-based approaches [1, 2]. Additionally, it would be beneficial to have a discussion on the reasons for focusing solely on high-correlation time instead of explicitly addressing prediction accuracy. Furthermore, given the frequent mention of Pearson correlation, a more detailed explanation of its relevance and its interpretation in all experimental results would be beneficial.


While it is mentioned that the U-net, which serves as the backbone for the current experiments, exhibited a smaller correlation time when using the pushforward trick from [2] in Figure 3, it is crucial to assess the PDE-Refiner method's performance in predicting long-term solutions for time-dependent PDEs across various models proposed in the field. In order to substantiate the claim that the PDE-Refiner method is superior, it is necessary to compare the accuracy of long roll-out predictions for PDE solutions using the proposed PDE-Refiner in this paper and other models employing different tricks in the context of time-dependent PDEs. Alternatively, if we assume that each network model has its own suitable trick and consider that the proposed PDE-Refiner aligns well with the U-net model, it would be appropriate to compare the accuracy of time-dependent PDE solution predictions between the U-net+PDE-Refiner approach in this paper and other models+tricks discussed in separate papers.

Lastly, I am curious about the necessity of associating PDE-Refiner with the Diffusion model. While both involve denoising, I wonder if it is necessary to establish a connection between the two (Section 3.1). While they share similarities in terms of denoising, it would be interesting to understand if linking them together is crucial to the novelty of the paper.

Overall, addressing these points and providing additional explanations would greatly enhance the clarity and comprehensibility of the paper.

[1] Boussif, Oussama, et al. "MAgnet: Mesh agnostic neural PDE solver." Advances in Neural Information Processing Systems 35 (2022): 31972-31985.

[2] Brandstetter, Johannes, Daniel Worrall, and Max Welling. "Message passing neural PDE solvers." arXiv preprint arXiv:2202.03376 (2022).

**Questions:**

* Do you think that PDE-Refiner is related to the neural operator method mentioned in [2]? If so, what are the similarities and differences between the autoregressive and temporal bundling approaches in [2] and the PDE-Refiner? Can these two models be applied to PDE-Refiner? What is the relationship between them?

* Are wavenumber and frequency ultimately the same thing?

* Looking at Figure 2, it seems that the model should involve the index $k$. How is this index incorporated? Since $\sigma_k$ will have different effects on the input as $k$ changes, is this index necessary as an input?

* In Equation (4), shouldn't the '$u(t)$' after 'NO' be $\hat{u}^k$ instead?

* As mentioned in line 203, the experiment was conducted with predictions made at intervals of $4\Delta t$. Would the results differ if the interval was shorter? How would they differ if it was longer? What range of variations is permissible?

* If more than two histories are used for History2 in Figure 3, how would the results change? Additionally, if predictions are made for multiple time steps, how would the results differ?

* The graph on the far right in Figure 4 is unclear in its meaning.

**Limitations:**

I am curious about the reason for presenting all experimental results in terms of correlation time rather than accuracy. It is possible that I may have misunderstood, but addressing the previous questions would provide a clearer understanding of the purpose and direction of the paper.

---

> ### Author Rebuttal · Authors · 2023-08-09
>
> Thank you for your constructive review and feedback. We provide below clarifications to your questions and concerns raised.
>
> **Clearer explanation of how Gaussian noise enables denoising across sigma values / Equation (4), shouldn't the $u(t)$ after 'NO' be $\hat{u}^k$ instead?**
>
> The refinement process of PDE-Refiner is trained by adding noise to ground truth data, $u(t)$ (Eq. (4)). This effectively removes information of frequencies below the amplitude of the noise, $\sigma_k$, which the model learns to recover. Frequencies above the amplitude are only minorly changed, such that the model learns to leave them mainly untouched. Thus, PDE-Refiner is trained to focus on different amplitude levels across the sigma values in the refinement process.
>
> During inference, we replace the ground truth data by the model's own prediction, $\hat{u}^k$, as depicted in Figure 2. Based on its training, PDE-Refiner will again focus on information below $\sigma_k$ and ignore potential errors in higher amplitudes. This is since the model was trained on ground-truth data, and learned to assume higher amplitudes to be correct.
>
> Your question raises a good point, since the mismatch of Figure 2 showing the inference and Equation (4) the training is slightly confusing. We will emphasize and clarify this difference.
>
> **Experimental results when applying PDE-Refiner to models like FNO and graph-based approaches**
>
> We agree and have provided FNO studies in Fig 13 and FNO ablations in Fig 14 in the submitted Appendix E.1. Further, in the rebuttal, we have added Dilated ResNet results (Figure 1). U-Nets, FNOs, and Dilated ResNets are the best performing models in both comprehensive overviews of Gupta \& Brandstetter [2022] and Stachenfeld et al. [2022]. On all methods, PDE-Refiner obtains a significant improvement. Regarding graph-based approaches: The proposed datasets are for regular gridded data. Non-regular data with graph-based approaches is left for future work.
>
> **Discussing reasons for focusing on high-correlation time instead of prediction accuracy**
>
> We are interested in how long the model stays accurate. This can be measured by high-correlation time, as in previous works, with thresholds of 0.8 and 0.9 indicating when the prediction starts to visually slightly differ from the ground truth. Alternative, we could check when the prediction exceeds a certain MSE loss, as shown in Fig 10 in Appendix D.1. Both measures come to a similar conclusion, while correlation is (1) invariant to scale and thus better comparable across PDEs, and (2) limited to [-1,1], such that it cannot be dominated by a single trajectory.
>
> **Comparison of PDE-Refiner and other models/tricks**
>
> We have done comparisons for different MSE training strategies (different history inputs, larger models, MSE ensembles), and for different alternative losses (pushforward, Sobolev, MNO, error correction, and error prediction) - for both U-Nets and FNOs, and added Dilated ResNets for the rebuttal. While some methods benefit more from FNOs than U-Nets (e.g. MNOs) or vice versa, they still stay all in the same performance range and are outperformed by PDE-Refiner. We are happy to add losses beyond those studied if suggested by the reviewer.
>
> **Associating PDE-Refiner with Diffusion crucial for novelty.**
>
> It is not strictly necessary, since PDE-Refiner has a different motivation than Diffusion models. We point out the connection for readers familiar with Diffusion models and clarify the differences.
>
> **PDE-Refiner is related to autoregressive modeling and temporal bundling of Brandstetter et al.? / If predictions are made for multiple steps, how would the results differ?**
>
> Temporal bundling (TB) can be easily applied to PDE-Refiner, we therefore denoise all outputs at the same time. Hence, we see TB as an orthogonal trick for obtaining stable rollouts. A possible combination of TB and PDE-Refiner would be to treat time as an extra dimension and thus refine over spatial and temporal frequency components, but be computationally expensive for long inputs. We add results for predicting up to 4 time steps in TB in rebuttal Figure 3. The performance slightly worsens due to the larger predicted time step.
>
> **Are wavenumber and frequency the same thing?**
>
> Correct! Wavenumber and frequency are defined up to some factor as the number of wavelengths per (unit) distance.
>
> **How is the k index incorporated?**
>
> The index $k$ is a conditional parameter passed to the model. We follow the standard implementation in Diffusion models of Ho et al. [2020] and the conditioning parameters in Gupta et al. [2022] by adding $k$ as input to each residual block in U-Nets, and condition the parameters in FNOs. A sketch of the conditioning on $k$ is provided in Fig 9 in the appendix.
>
> **Would results differ if the interval was shorter/longer than $4\Delta t$?**
>
> See Appendix E.2 for a detailed ablation study. Smaller time steps generally give better results, but are more expensive to roll out. Larger time steps lead to suboptimal results and more difficult generalization.
>
> **If more than two histories are used, how would the results change?**
>
> See Appendix E.3. We found more history gets even worse, especially for small time steps.
>
> **Graph on the right in Fig 4.**
>
> It shows the average frequency band of white noise that we sample at the different refinement steps, which has different orders of magnitude. In other words, it is FFT$(\sigma\_k\cdot\epsilon\_k)$. It gives a relation between the error and noise per step in the refinement process. We will clarify it in the paper.
>
> ---
>
> [Gupta \& Brandstetter, 2022] Gupta, Jayesh K., and Johannes Brandstetter. Towards multi-spatiotemporal-scale generalized pde modeling. arXiv preprint arXiv:2209.15616 (2022).
>
> [Ho et al., 2020] Ho, Jonathan, et al. Denoising diffusion probabilistic models. NeurIPS (2020).
>
> [Stachenfeld et al., 2022] Stachenfeld, Kimberly, et al. Learned coarse models for efficient turbulence simulation. ICLR (2022).

---

> > ### Comment · Reviewer_K6r8 · 2023-08-14
> > **Reply**
> >
> > Thank you for your detailed responses to my questions. I appreciate your efforts in clarifying the points that were causing confusion for me. I now have a much better understanding of most of the aspects that were unclear.
> >
> > There is one more point that I have become curious about. In both Figure 2 of the main text and the Pseudocode in Appendix - Section C, it appears that the parameter "k" is input to the self.Neural operator whether it is 0 or non-zero. Does this mean that the neural operator (depicted as a green circular node) outputs $\hat{u}^1$ (when k=0) and $\hat{\epsilon}^k$ (when k is not 0)? Considering that the scales of $u$ and $\epsilon$ are different, I wonder if it is feasible for a single model to generate both outputs simultaneously. I am curious if there are any concerns regarding this arrangement.
> >
> > I have also read your response to question
> >
> > > Associating PDE-Refiner with Diffusion crucial for novelty.
> >
> > This leads me to wonder about the necessity and purpose of Section 3.1. Is it intended to explain something entirely distinct from the diffusion model, as you mentioned? Could you elaborate a bit more on why Equation (5) was included and provide further insight into the reasons behind its formulation? Your clarification on these points would be greatly appreciated.

---

> > > ### Author Response · Authors · 2023-08-15
> > > **Follow-up response**
> > >
> > > Thank you for your going through the rebuttal in detail! We are happy to see that our rebuttal clarified most points. We will clarify your remaining questions below.
> > >
> > > **Does this mean that the neural operator (depicted as a green circular node) outputs $\hat{u}^1$ (when $k=0$) and $\hat{\epsilon}^k$ (when k is not 0)? Considering that the scales of $\hat{u}^1$ and $\hat{\epsilon}^k$ are different, I wonder if it is feasible for a single model to generate both outputs simultaneously. I am curious if there are any concerns regarding this arrangement.**
> > >
> > > Indeed, we are using the same neural network for predicting $\hat{u}^1$ when $k=0$, and $\hat{\epsilon}^k$ when $k>0$. As a clarification, all prediction targets ($\hat{u}^1$ and $\hat{\epsilon}^k$ for any $k>0$) have a standard deviation of around 1, thus being of the same scale. Note that, similar to diffusion models [Ho et al., 2020], we predict the normalized noise $\hat{\epsilon}^k\sim\mathcal{N}(0,1)$ instead of the weighted noise $\sigma_k\hat{\epsilon}^k$ which is added to the signal, since this optimizes much better.
> > >
> > > Still, you raise a good point of whether separating the two operators would be beneficial. We performed experiments on that during the development of the method and found that there was no performance gain in having separate neural operators. This is because the refinement process doesn't need the first prediction to be perfect, just ideally have a lower error than the noise we add. The intuition also aligns with the v-prediction in Diffusion models [Salisman and Ho, 2022] which has a similar combination of signal and noise prediction. Further, having separate neural operators would increase training cost, and there are certainly features to be shared across the refinement steps (e.g. analyze the previous time step). This makes our current setup with a single neural operator the more efficient and preferable option in practice. We will try to mention these insights in the main text.
> > >
> > > **[Section 3.1] Is it intended to explain something entirely distinct from the diffusion model, as you mentioned? Could you elaborate a bit more on why Equation (5) was included and provide further insight into the reasons behind its formulation?**
> > >
> > > Section 3.1 is not intended to explain something entirely distinct from Diffusion models, but highlight the connections and differences between PDE-Refiner and Diffusion models. Diffusion models are at the moment one of the most popular models deploying a denoising process, as PDE-Refiner. Thus, we expect readers familiar with Diffusion models to wonder about the relation between the models, which we clarify in Section 3.1. Furthermore, Section 3.1 gives such readers an easier access to connect their knowledge on Diffusion models to PDE-Refiner, for instance in what optimizations and architectures may work especially well. For example, in our response to your previous question above, we mentioned how the intuition of why certain optimizations work in PDE-Refiner relate to the well-studied domain of Diffusion models.
> > >
> > > Equation 5 was mainly included for introducing Diffusion models, and how one could apply Diffusion models to the autoregressive prediction in neural PDEs (line 146-148). Based on the relation between PDE-Refiner and the probabilistic nature of Diffusion models shown in Eq. 5, we argue for why PDE-Refiner can obtain accurate uncertainty estimates. Still, these arguments could also be made without explicitly having Eq. 5, if you would suggest that.
> > >
> > >
> > > Thank you again for your effort in reviewing the paper! Please let us know if you have any further questions or concerns.
> > >
> > > ---
> > >
> > > [Ho et al., 2020] Ho, Jonathan, et al. Denoising diffusion probabilistic models. NeurIPS (2020).
> > >
> > > [Salisman and Ho, 2022] Salimans, Tim, and Jonathan Ho. "Progressive distillation for fast sampling of diffusion models." arXiv preprint arXiv:2202.00512 (2022).

---

> > > > ### Comment · Reviewer_K6r8 · 2023-08-17
> > > > **Reply2**
> > > >
> > > > Thanks for the explanation. I have no further question and I'm glad to raise my score by one.

---

### Author Rebuttal · Authors · 2023-08-09

We would like to thank all reviewers for their constructive and valuable feedback.

We are encouraged to see that reviewers highlight this study as interesting (YYxA, JgV1, xf2d), the good presentation of the paper (YYxA, xf2d), and the thorough ablation studies (K6r8, JgV1, xf2d). For example, reviewer xf2d describes the paper as a "high-quality paper that is well-polished and has substantial strengths in terms of contributions, novelty, quality and presentation". Reviewer JgV1 mentions "this paper presents an interesting finding", and reviewer YYxA that our analysis "can inspire future research". Finally, reviewer K6r8 "appreciate[s] the thorough review of the paper".

We welcome the opportunity to clarify all concerns and questions raised. We would first like to point out that our submitted appendix contains a rich set of ablation studies and further experimental results, for which the references in the main paper may have been easily missed in its current version. We will point out the appendix references more clearly in the final paper where needed, and link here the sections that address reviewer's concerns, such as:

- applications to other models (Appendix E.1), where we have added FNO studies (matching the presented ones in the main paper) in Fig 13 and FNO ablations in Fig 14 (reviewer K6r8, JgV1);
- alternative study to high-correlation time (Appendix D.1), in which we check when the prediction exceeds certain loss as shown in Fig 10 (reviewer K6r8);
- ablations on different time intervals and different input histories in Appendix E.2 and Appendix E.3 (reviewer K6r8);
- and repeating the frequency analysis for the 2D Kolmogorov flow in Appendix E.5 (reviewer YYxA).

Based on the reviewer's suggestion, we provide additional empirical results in the submitted 1-page rebuttal PDF:

- Figure 1: results on another SOTA neural operator, Dilated ResNets [Stachenfeld et al., 2022]. PDE-Refiner significantly improves performance on this operator as well (reviwer JgV1, K6r8);
- Figure 2: showing the consistency of our findings across threshold values on the high-correlation time by plotting the correlation of different methods over rollout time. The trend stays the same for various thresholds (reviewer xf2d);
- Figure 3: predicting multiple time steps at once. The performance is limited by the same generalization challenges as predicting at large time steps (reviewer K6r8);
- Figure 4: applying PDE-Refiner to smaller datasets. The experiments highlight PDE-Refiner's data efficiency over standard neural operators, in which the denoising objective acts as augmentation (reviewer JgV1);
- Table 1: the GPU memory required by PDE-Refiner only slightly differs from standard operator training, increasing it by less than 1\% for both training and inference (reviewer JgV1).

Due to the limited rebuttal time, we show some results for a single seed only, whereby the general trends were always well outside the standard deviation ranges. For the final paper, we will repeat the experiments over multiple seeds, as done for the other experiments in the main paper.

For further clarification, please refer to our individual responses to each reviewer. If any concerns, unclarities or questions of the reviewers remain, we would be happy to answer them and elaborate on our previous responses during the discussion phase.

---

[Stachenfeld et al., 2022] Stachenfeld, Kimberly, et al. Learned coarse models for efficient turbulence simulation. ICLR (2022).

---

### Decision · Program_Chairs · 2023-09-21

**Decision:**

Accept (spotlight)

**Comment:**

The paper connects a multi-step prediction for PDEs with diffusion models. This is a fresh view on the subject, with interesting algorithmic ideas and convincing numerical results. The reviewers have valued the results as well.